# Agent-ScanKit: Unraveling Memory and Reasoning of Multimodal Agents via Sensitivity Perturbations

## Abstract

Although numerous strategies have recently been proposed to enhance the autonomous interaction capabilities of multimodal agents in graphical user interface (GUI), their reliability remains limited when faced with complex or out-of-domain tasks. This raises a fundamental question: Are existing multimodal agents reasoning spuriously? In this paper, we propose **Agent-ScanKit**, a systematic probing framework to unravel the memory and reasoning capabilities of multimodal agents under controlled perturbations. Specifically, we introduce three orthogonal probing paradigms: visual-guided, text-guided, and structure-guided, each designed to quantify the contributions of memorization and reasoning without requiring access to model internals. In five publicly available GUI benchmarks involving 18 multimodal agents, the results demonstrate that mechanical memorization often outweighs systematic reasoning. Most of the models function predominantly as retrievers of training-aligned knowledge, exhibiting limited generalization. Our findings underscore the necessity of robust reasoning modeling for multimodal agents in real-world scenarios, offering valuable insights toward the development of reliable multimodal agents. Our code is available at anonymous.

## 1 Introduction

With recent advances in multimodal large language models (MLLMs) (Hurst et al., 2024; Team, 2025; Shen et al., 2025a), building multimodal agents has become more straightforward and generalizable, particularly in graphical user interfaces (GUIs). These agents promise broad task automation on mobile and desktop devices (Wang et al., 2024b; Zhang et al., 2024a). Through environmental perception (e.g., screen) (Ma et al., 2024), MLLM-based GUI agents predict the subsequent action based on a specific goal. As shown in Figure 1, recent work advances grounding (Wu et al., 2024b; Qin et al., 2025; Zhou et al., 2025), planning (Zhang et al., 2024d; Wu et al., 2025b), reflection (Lu et al., 2025; Luo et al., 2025b; Liu et al., 2025d), and adaptation (Bai et al., 2024; Wang et al., 2024c) through continue pretraining (CPT), supervised fine-tuning (SFT), and reinforcement learning (RL). Notable RL variants include Direct Preference Optimization (DPO) (Rafailov et al., 2023) and Group Relative Policy Optimization (GRPO) (Shao et al., 2024).

However, existing open-source multimodal agents still exhibit poor reliability when faced with complex or out-of-domain (OOD) GUI tasks (Wu et al., 2025c; Liu et al., 2025b; Guo et al., 2025b). Related studies further suggest that the so-called "reasoning" ability of LLMs often reduces to sophisticated pattern matching (Mirzadeh et al., 2024) or even rote memorization of training data (Carlini et al., 2021; Hartmann et al., 2023). Therefore, we conduct a systematic analysis and identify three core contributors to unreliability.

First, the inherently unbounded nature of visual and textual spaces results in potential visual and textual-oriented *memory biases*, which decrease the accuracy of the prediction and directly undermine the success rates of the tasks. Second, prior research has focused primarily on learning within these two spaces, while overlooking the optimization of state and reflection action, thus introducing varying degrees of action-based *memory shortcuts*. Third, *domain sensitivity* further limits the agent's ability to generalize across tasks and environments. Although many models report stepwise accuracy (SR) above 80% and task success rates above 40% on benchmarks such as AITZ (Zhang et al., 2024c) and

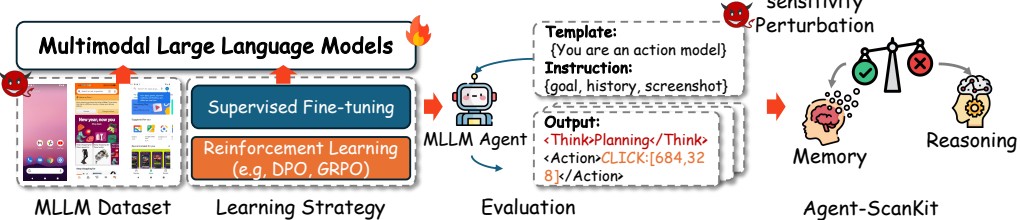

Figure 1: Pipeline of existing multimodal agents for GUI tasks.

AndroidControl (Li et al., 2024a) (Table 6), a significant performance drop when these models are evaluated on long-horizon tasks or cross-platform scenarios (Table 7). These observations motivate a key research question: *Memory or reasoning: what drives multimodal agents?*

In this paper, we propose **Agent-ScanKit**, a systematic probing toolkit to unravel memory and reasoning capabilities in multimodal agents by sensitivity perturbation. Specifically, we introduce three orthogonal probing paradigms:

(i) In the *visual-guided level*, we use object masking and editing to test whether grounding relies on memorization, while a zooming strategy quantifies reasoning under local visual changes.

(ii) In the *text-guided level*, we adopt atomic instruction masking in token-level and substitution in sentence-level, aiming to probe memory and reasoning in textual modal.

(iii) In the *structure-level*, we probe that specific status and reflection actions are memory shortcuts, or reasoning caused by reflection.

Each was designed to isolate and quantify the contributions of memorization and reasoning without requiring access to the model's internals. In five publicly available GUI benchmarks involving 18 agents, the results reveal that existing multimodal agents exhibit over-memorization in three probing strategies. Concretely, these agents tend to construct complex, brittle mappings between inputs and outputs, acting more as retrievers of training-aligned knowledge than as genuine reasoners. Furthermore, RL-based methods combined with the chain-of-thought (CoT) mechanism have some reasoning capabilities on language modal-side, enabling the competence extrapolation and environmental adaptability. These findings clearly define genuine reasoning mechanisms within multimodal agents for building more reliable and general-purpose AI assistants. Our contributions can be summarized as follows:

(i) We conduct a comprehensive evaluation of 18 open-source GUI agents on 5 benchmarks, revealing two central challenges: the infinite predictive space and finite generalization.

(ii) We present Agent-ScanKit, a systematic probing toolkit, which provides a unified analysis across visual, textual, and structural dimensions through sensitivity perturbations, enabling quantitative assessment of memory and reasoning in multimodal agents.

(iii) We show that existing multimodal agents often display spurious reasoning behaviors driven by over-memorization. Although RL and CoT-augmented strategies have facilitated progress in GUI tasks, substantial room for improvement remains.

## 2 RELATED WORKS

This section reviews two lines of research that form the basis of this work: (i) multimodal agents for GUI interaction, and (ii) internal mechanisms for memory and reasoning.

### 2.1 MLLM-BASED GUI AGENTS

The rise of MLLMs (Team, 2025; Zhang et al., 2024a) has a significant shift in GUI automation. By perceiving UI states (e.g., screenshots) and performing atomic actions like clicks and typing, multimodal agents enable more flexible, human-like interactions across platforms, such as Desktop (Niu et al., 2024; Zhang et al., 2024b; Wu et al., 2024a), Web (Gur et al., 2023; Zheng et al., 2024; Ma et al., 2023; Shen et al., 2025b), and Mobile (Zhang et al., 2025b). Following SPA-Bench (Chen

et al., 2024a) and RiOS-World (Yang et al., 2025), existing GUI agents can be categorized into two paradigms: agentic workflows and agent-as-a-model.

The former is framework-based that adopts prompt learning on a proprietary model and leverages the power of MLLMs (e.g., GPT-4o and Claude 3.5 Sonnet) to build environment perception (Zhang et al., 2025b; Li et al., 2024b), task planning (Guo et al., 2025b), decision reflection (Rawles et al., 2024; Liu et al., 2025c), memory persistence (Dai et al., 2025; Jiang et al., 2025), and multi-agent collaboration (Wang et al., 2024a; 2025b;b; Zhang et al., 2024b; Khaokaew et al., 2024). However, practitioners have raised concerns about privacy leakage, the cost of API usage, and latency during inference on real-world devices. In addition, task performance is generally poorer compared to the latter. In contrast, the agent-as-a-model centers on building native agent models. By customizing MLLMs through CPT, SFT, and RL for agentic tasks, workflow knowledge is embedded directly into the model itself. This enables capabilities such as grounding enhancements (Qin et al., 2025; Zhou et al., 2025; Zhang et al., 2025e; Wu et al., 2025e), planning (Zhang et al., 2024d; Wu et al., 2025b), reflection (Zhang et al., 2024c; Luo et al., 2025b; Lu et al., 2025; Wanyan et al., 2025; Wu et al., 2025a), environmental adaptation (Bai et al., 2024; Wang et al., 2024c; Xie et al., 2025), experience replay (Liu et al., 2025a; Zhang et al., 2025a) and reliability (Ma et al., 2024; Cheng et al., 2025a;b; Wu et al., 2025d). Nevertheless, these models struggle on complex or OOD tasks, motivating us to quantify their memory and reasoning capabilities for deeper insight into their execution mechanisms.

## 2.2 Memory vs. Reasoning

Recently, two perspectives on the execution mechanism of LLMs have emerged: reasoning vs. memory. The former has been demonstrated in tasks such as mathematics (Wang et al., 2025c; Luo et al., 2025a) and QA (Chen et al., 2025; Guo et al., 2025a), where LLMs appear to provide correct answers by CoT. However, several studies have shown that the purported "reasoning" ability of LLMs is largely attributable to sophisticated pattern matching (Mirzadeh et al., 2024; Carlini et al., 2021; Hartmann et al., 2023). They also investigated the formation and contribution of memories (Speicher et al., 2024; Dankers & Titov, 2024), and demonstrated that such mechanisms are effective primarily on simple tasks (Li et al., 2024c; Jin et al., 2025). Further studies highlight the importance of detecting and disentangling LLM memorization (Djiré et al., 2025; Jin et al., 2024), as well as exploring how such mechanisms can be systematically measured (Schwarzschild et al., 2024). Thus, given the context of poor reliability of multimodal agents in GUI tasks, the quantification of memory and reasoning capabilities becomes critical.

## 3 Challenge of Multimodal Agents

In this section, we first formalize multimodal GUI agents and then conduct a benchmark evaluation to introduce two limitations. For completeness, we report detailed results in Appendix B.1.

### 3.1 Problem Statement

**MLLM-based GUI Agents.** Following prior works (Wu et al., 2025b; Wang et al., 2025a), we formalize GUI agentic tasks as a goal-driven partially observable Markov decision process (POMDP), defined by the tuple $\mathcal{M} = (G, \mathcal{S}, \mathcal{A}, \mathcal{T}, \mathcal{R}, \mathcal{H})$. Here, $G$ denotes the goal space, $\mathcal{S}$ the perceptual state space (e.g., screenshots and supplementary data (Cheng et al., 2025a)), $\mathcal{A}$ the action space (Appendix A.1), $\mathcal{T} : \mathcal{S} \times \mathcal{A} \rightarrow \mathcal{S}$ the transition function, $\mathcal{R} : \mathcal{S} \times \mathcal{A} \rightarrow \mathbb{R}$ the bounded reward function, typically positive upon task completion, and $\mathcal{H}$ the maximum action steps for a goal.

Given a user goal $g \in G$, the agent observes the environment state $s_t$ at time $t$ and predicts an action $a_t \in \mathcal{A}$ through a structured reasoning process. This reasoning process may involve a CoT $r_t$, which allows the agent to interpret its observations and refine its decisions step by step. Executing $a_t$ yields the next state $s_{t+1}$, and the trajectory $(s_{1:n}, r_{1:n}, a_{1:n})$ constitutes an episode associated with $g$, formalized as $E = (g, \{s_t, r_t, a_t\}_{t=1}^n)$.

As just discussed, the development of multimodal agents for GUI tasks is generally unified under a three-stage training framework (Tang et al., 2025), comprising perception enhancement through CPT, behavioral imitation via SFT, and generalization with RL, further reinforced by data and model scaling laws. Despite this progress, their capabilities remain constrained: agents often rely on visual-textual

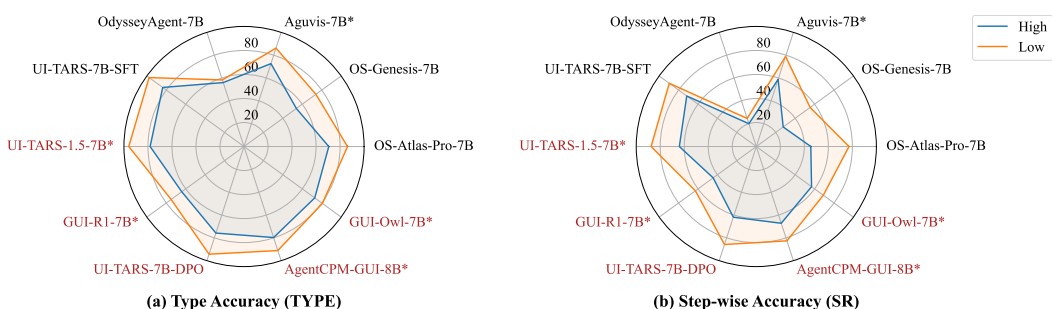

(a) Type Accuracy (TYPE)  (b) Step-wise Accuracy (SR)

Figure 2: Comparative performance of 7∼8B multimodal agents on two evaluation metrics in GUI tasks. RL-based models are highlighted in red, while reasoning-enabled models are marked with "*". Low-level provides atomic instructions based on queries, whereas high-level only offers the query.

rule matching rather than understanding the operational logic of GUl. We refer to this phenomenon as memory-driven spurious reasoning. To this end, we first conduct a comprehensive evaluation of existing agents and identify two key challenges: (i) infinite predictive space; (ii) finite generalization.

## 3.2 FAILS DUE TO INFINITE PREDICTIVE SPACE

In GUI tasks, multimodal agents require not only accurate predictions of coordinates, and text within the TYPE action, but also determine the task status and promptly engage in reflection. However, existing data distribution and algorithm exhabit over-optimization on the infinite coordinate space and vocabulary space. Therefore, we first investigate the detrimental effects of overemphasizing these two predictive spaces on multimodal GUI agents.

To begin with, we present the performance for 10 multimodal agents with 7 to 8B scale under the AndroidControl benchmark (Li et al., 2024a), as shown in Figure 2. Fine-grained action-type accuracy under the low-level setting is reported in the Appendix B.1.1. Overall, existing muiltimodal agents perform relatively robustly on actions in coordinate-based (e.g., CLICK) and vocabulary-based (e.g., TYPE). However, over-reliance on foundational data and perception-enhancement strategies may cause models to exhibit memory bias rather than genuinely comprehend the operational logic of graphical user interfaces. In addition, depending solely on suboptimal data distributions without stronger algorithmic support leaves considerable room for improvement in vocabulary learning. Most importantly, the performance of reflection actions (e.g. PRESSHOME and PRESSBACK) and state actions (e.g., WAIT and COMPLETE) poorly executed, forcing the model to learn shortcuts, thereby achieving local optima. These side-effects further make SR accuracy declines sharply. Furthermore, the strong performance boost from atomic (low-level) instructions suggests that agents may engage in a text-guided reasoning process that does not rely on visual signals, especially for reflective or state-oriented actions. Therefore, detecting whether multimodal agents are relying on rote memorization or genuine reasoning is of paramount importance.

## 3.3 FAILS DUE TO FINITE GENERALIZATION

We divide the finite generalization into task and environment categories. For GUI agents to execute tasks reliably, they also need to satisfy summarization and reasoning beyond their training data, thereby extending the generalization. We thus investigate whether finite generalization also underlie their inferior performance through task success rates.

As shown in Figure 3, we evaluate 18 models on 5 benchmark datasets. Our results show that early SFT-based agents consistently underperform, underscoring the limitations of relying on a single training paradigm. In contrast, later models achieve substantial gains as data coverage, model capacity, and training sophistication increase. The improvement is most evident on datasets such as AndroidControl and AITZ, which likely reflects exposure to similar scenarios during training. However, these gains do not generalize. Their performance drops markedly on long-horizon tasks (e.g., GUI Odyssey) and platform-across (e.g., GUI Act-Mobile, Web). This pattern highlights the intrinsic limits of current generalization. Although scaling and strategy optimization deliver clear benefits, agents remain tightly coupled to the distributions seen in training.

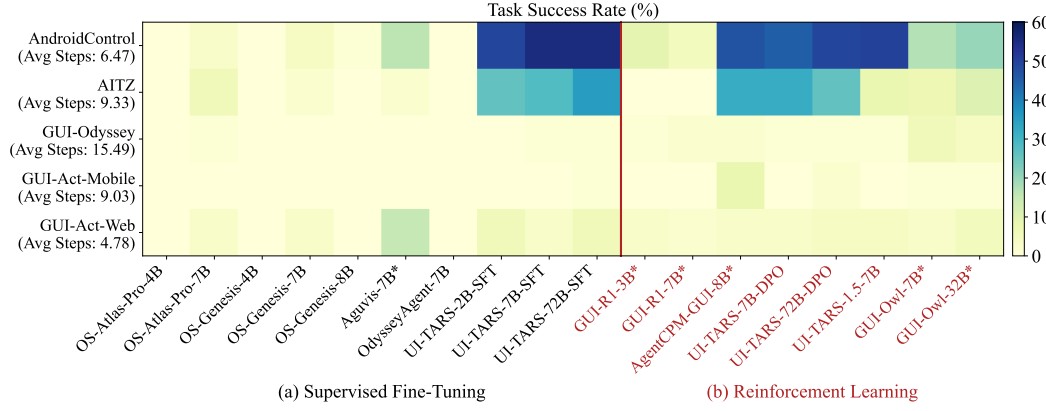

(a) Supervised Fine-Tuning          (b) Reinforcement Learning

Figure 3: Task success rates for multimodal agents across five datasets. Models on the x-axis are grouped by training paradigm. The y-axis lists datasets, with parentheses indicating each dataset's average interaction lengths (Avg Steps). "*" denotes models providing CoT for action reasoning.

Building on these results, we further highlight two complementary findings. (i) Although RL is commonly regarded as a pathway to generalization, we observe that SFT-based models outperform their RL counterparts. (ii) CoT-augmented agents (e.g., AgentCPM-GUI) not only provide decision interpretability but also match the performance of the strongest non-reasoning models. In contrast to the conclusions of (Zhang et al., 2025d), our results suggest that CoT remains essential to advance multimodal agents, although still imperfect. To this end, we provide a quantitative analysis of memory and reasoning to explain the limited generalization of multimodal agents.

## 4 AGENT-SCANKIT

Based on the observations in Section 3, we propose **Agent-ScanKit**, a probing toolkit that systematically quantifies the memory and reasoning capabilities of multimodal agents under controlled input perturbations. As illustrated in Figure 4, Agent-ScanKit incorporates three orthogonal probing paradigms: (i) visual-guided, (ii) text-guided, and (iii) structure-guided.

Following the POMDP formulation of GUI tasks, we extend the perceptual state space $\mathcal{S}$ with perturbation operators $\mathcal{P}$, forming a perturbed POMDP:

$$\mathcal{M}_p = (G, \mathcal{S}, \mathcal{A}, \mathcal{T}, \mathcal{R}, \mathcal{H}, \mathcal{P}), \tag{1}$$

where $\mathcal{P} : S \rightarrow S$ modifies the observed state $s_t$ in time step $t$. Given a goal $g \in G$, the agent receives perturbed observations $s'_t = \mathcal{P}(s_t)$, and selects action according to:

$$a_t \sim \pi(a_t|s'_t, g). \tag{2}$$

By contrasting agent performance under perturbed versus unperturbed conditions, we quantify perturbation sensitivity as:

$$\begin{aligned}
\Delta_P &= \mathbb{E}_{(g,s_t)}[Acc(\pi(s_t, g)) - Acc(\pi(\mathcal{P}(s_t), g))] \\
&= 1 - \mathbb{E}_{(g,s_t)}[Acc(\pi(\mathcal{P}(s_t), g))],
\end{aligned} \tag{3}$$

where $Acc(\pi(s_t, g))$ is a solvable subset of the agent. Similarly, goal perturbations $\mathcal{P} : G \rightarrow G$ is used to probe text- and structure-guided mechanisms.

### 4.1 VISUAL-GUIDED PROBING

In the visual-guided level, we hypothetical existing agents often exploit spatial priors (e.g., "confirm buttons usually appear at the bottom-right") rather than reasoning over screen content. To quantify such visual memory biases, we propose visual-guided perturbations: object masking and editing by obscuring or removing targets, while zoom-in introduces an OOD-like state to evaluate reasoning beyond rote memorization (Figure 4).

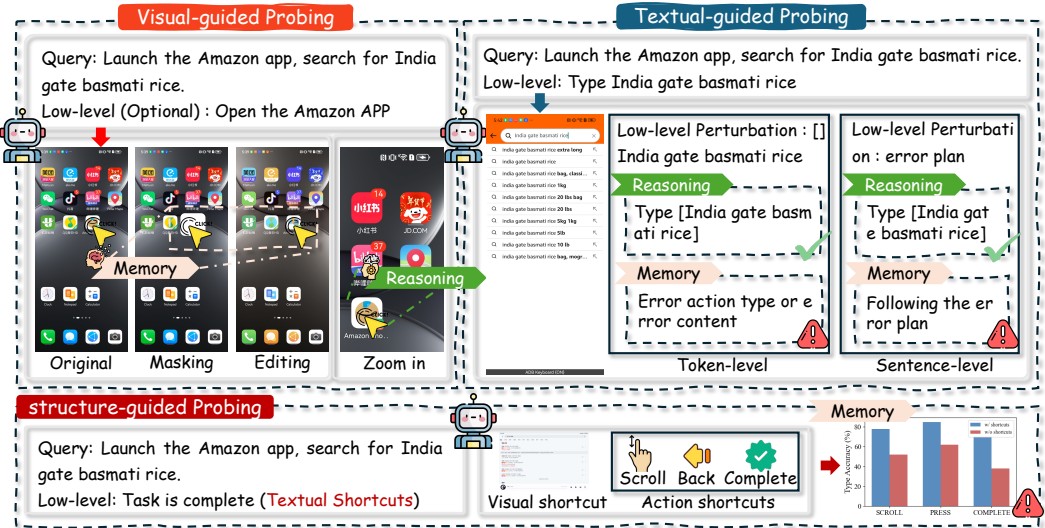

Figure 4: Overview of the Agent-ScanKit framework that applies controlled visual, textual, and structural perturbations to multimodal agents, revealing how memory and true reasoning interact.

**Object Masking & Editing.** We introduce two operators that directly alter the ground-truth target element $e^* \subseteq s_t$:

$$\mathcal{P}\left(s_t, e^*\right): \quad s_t'(x, y) = \begin{cases} 0, & (x, y) \in \Omega\left(e^*\right) \\ s_t(x, y), & \text{otherwise} \end{cases} \tag{4}$$

where $\Omega\left(e^*\right)$ denotes the spatial region of $e^*$. Object masking can remove perceptual evidence of $e^*$, evaluating whether the agent relies on memorized spatial priors.

To probe the depth of spatial memory, we employ object editing, a perturbation that eliminates target elements and reconstructs them with interpolation over neighboring pixels:

$$\mathcal{P}\left(s_t, e^*\right): \quad s_t'(x, y) = \begin{cases} \delta\left(s_t(x, y)\right), & (x, y) \in \Omega\left(e^*\right) \\ s_t(x, y), & \text{otherwise ,} \end{cases} \tag{5}$$

where $\delta(\cdot)$ is the image-editing algorithm. If the performance remains stable under perturbations, i.e., when $\Delta_{\mathcal{P}}$ is small, it indicates that the agent's is primarily driven by memory retrieval rather than genuine reasoning, and vice versa.

**Zoom-in.** To evaluate reasoning, we define the zoom operator:

$$\mathcal{P}\left(s_t, q^*\right): \quad s_t' = \text{Crop}\left(s_t, q^*\right), \tag{6}$$

where $\{q_1, q_2, q_3, q_4\}$ partition the UI into quadrants, and $q^*$ is the quadrant containing the target $e^*$. Notably, zoom-in removes global layout information while preserving local fidelity. If the agent successfully identifies $e^*$ under $\mathcal{P}$, i.e., $\Delta_{\mathcal{P}}$ is small, it demonstrates contextual reasoning, vice versa.

Furthermore, we introduce two complementary metrics to quantify the model's behavioral responses. The first, visual memory consistency (VMC), measures how closely the model's predictions under masking/editing align with those from the original image. The second, reflection score (RS), reflects whether the agent successfully triggers appropriate reflective actions (e.g., SCROLL, PRESSBACK, PRESSHOME) or status-related actions (e.g., COMPLETE, WAIT) when presented with perturbed inputs. Additional details are provided in Appendix A.5.

### 4.2 TEXT-GUIDED PROBING

Multimodal agents operate in a joint visual–textual space, where the textual goal $g \in G$ guides perception and decision-making. Yet, agents are sub-optimized in navigating the infinite vocabulary space. It is unclear whether this stems from memorizing atomic instructions or from an inability to reason over user queries. To probe this distinction, we introduce text-guided perturbations at both the token and sentence levels in the low-level setting.

**Token-level.** We hypothesize that verbs or objects in the atomic instruction are pivotal for memory-driven behavior under the given text $g$. Accordingly, we modify the instruction as $g' = g \setminus \{w_i\}$. If $w_i$ is correctly inferred, $\Delta_{\mathcal{P}}$ should remain sufficiently small, and vice versa.

**Sentence-level.** We hypothesize that atomic instructions are central to memory-driven text processing. Thus, we substitute the instructions by setting $g' = \tilde{g}$ that $\tilde{g} \neq g$. If $g'$ can be disregarded, $\Delta_{\mathcal{P}}$ should remain small, and vice versa.

### 4.3 STRUCTURE-GUIDED PROBING

As discussed in Section 3, multimodal agents exhibit inherent optimization biases, which manifest as suboptimal reflection and state–action decisions. We conjecture that such behaviors arise from two systematic memory shortcuts internalized by the model: visual shortcuts and action shortcuts.

**Visual Shortcuts.** We define a visual shortcut as the model relying solely on the current screen $s_t$ for reflection or state-action decisions. Formally, if retaining only the current screen $s_t$ allows the agent to generate the action $a_t = \pi(a_t \mid s_t)$ with minimal $\Delta_{\mathcal{P}}$, then the agent's behavior is dominated by visual shortcuts; otherwise, no such shortcut exists.

**Action Shortcuts.** We define a action shortcut as the case where the model relies exclusively on the atomic instruction $g$ for reflection or state decision-making. Formally, if retaining only $g$ enables the agent to generate the action $a_t = \pi(a_t \mid g)$ with minimal $\Delta_{\mathcal{P}}$, then the agent's behavior is dominated by action shortcuts; otherwise, no such shortcut exists.

## 5 EXPERIMENTS

### 5.1 EXPERIMENT SETUPS

**MLLM-based GUI Agents.** We evaluate 18 representative open-source models developed by 8 institutions, spanning diverse architectural and training paradigms. These include the OS-Atlas series (Wu et al., 2024b), the OS-Genesis series (Sun et al., 2024), the UI-TARS series (Qin et al., 2025), Aguvis-7B (Xu et al., 2025), OdysseyAgent-7B (Lu et al., 2024), the GUI-R1 series (Luo et al., 2025b), the Mobile-Agent series (Ye et al., 2025), and AgentCPM-GUI-8B (Zhang et al., 2025e). Moreover, we extend evaluation to 3 closed-source models, including GPT-4o, GLM-4.5V, and Claud-4-Sonnet. More details appear in the Appendix A.2.

**Evaluation Benchmarks.** We evaluate five representative GUI benchmarks across three different platforms: AndroidControl (Li et al., 2024a), AITZ (Zhang et al., 2024c), GUI-Odyssey (Lu et al., 2024), and GUI-Act-Mobile (Chen et al., 2024b) for mobile agents; GUI-Act-Web and OmniAct-Web (Kapoor et al., 2024) for web agents; and OmniAct-Desktop for Windows environments. More details appear in the Appendix A.3.

**Settings.** In our benchmark evaluation, we report model performance under high-level and low-level settings. For visual-guided probing, we evaluate on the grounded samples from Table 6. For the text-guided probing, we use samples that require textual input from Table 6. For structure-guided probing, we focus on reflective actions and state actions. All evaluations employ the official open-source prompts and inference parameters. Unless otherwise specified, each agent is conducted under the low-level setting using samples with 100% SR accuracy. More details appear in the Appendix A.4.

**Metrics.** We use four metrics to evaluate all multimodal agents: accuracy of action-type prediction (Type), accuracy of coordinate prediction (Grounding), step-wise success rate (SR), and task success rate (TSR). Unless otherwise specified, we report $\Delta P_{\text{Type}}$ and $\Delta P_{\text{SR}}$ in probing experiments. In addition, we introduce two complementary metrics in visual-guided probing: visual memory consistency (VMC) and reflection score (RS). More details appear in the Appendix A.5.

### 5.2 MAIN RESULTS AND ANALYSIS

We present key findings at three levels of probing: visual-guided (Section 5.2.1), text-guided (Section 5.2.2) and structure-guided (Section 5.2.3), each revealing a distinct balance between memory and reasoning in multimodal GUI agents.

Table 1: Visual-Guided Probing on 7∼8B multimodal agents in GUI tasks. Memory is evaluated through object masking/editing, whereas reasoning is assessed via zoom-in. The masking and editing ratio, and the distance threshold of VMC are set to 50 pixels. $\Delta P_{\text{Type}} \downarrow$ and $\Delta P_{\text{SR}} \downarrow$ denote changes in prediction performance, where the direction of the arrows indicates greater reasoning ability.

| GUI Agents | Object Masking | | | | Object Editing | | | | Zoom-in | | | |
|---|---|---|---|---|---|---|---|---|---|---|---|---|
| | $\Delta P_{\text{Type}} \uparrow$ | $\Delta P_{\text{SR}} \uparrow$ | VMC↓ | RS↑ | $\Delta P_{\text{Type}} \uparrow$ | $\Delta P_{\text{SR}} \uparrow$ | VMC↓ | RS↑ | $\Delta P_{\text{Type}} \downarrow$ | $\Delta P_{\text{SR}} \downarrow$ | VMC↓ | RS↓ |
| **Supervised Fine-Tuning** | | | | | | | | | | | | |
| OS-ATLAS-Pro-7B | 13.0 | 58.9 | 14.5 | 11.6 | 8.50 | 42.6 | 39.1 | 7.52 | 13.6 | 40.5 | 0.72 | 12.3 |
| OS-Genesis-7B | 1.11 | 20.4 | 94.7 | 0.23 | 1.09 | 20.7 | 93.4 | 0.34 | 2.90 | 98.8 | 86.0 | 1.61 |
| OS-Genesis-8B | 0.20 | 7.40 | 98.8 | 0.12 | 0.40 | 7.70 | 98.7 | 0.14 | 1.80 | 96.2 | 95.6 | 0.09 |
| OdysseyAgent-7B | 1.90 | 37.9 | 57.6 | 1.45 | 1.50 | 33.8 | 63.6 | 0.96 | 3.90 | 62.0 | 5.69 | 1.92 |
| Aguvis-7B | 0.10 | 13.3 | 99.5 | 0.02 | 0.50 | 1.80 | 97.2 | 0.03 | 5.40 | 47.0 | 0.65 | 1.64 |
| UI-TARS-SFT-7B | 10.7 | 37.8 | 53.3 | 8.05 | 7.20 | 33.8 | 60.9 | 5.12 | 12.1 | 36.5 | 1.43 | 8.66 |
| **Reinforcement Learning** | | | | | | | | | | | | |
| GUI-R1-7B | 13.1 | 52.4 | 43.4 | 12.0 | 9.20 | 37.0 | 54.2 | 8.66 | 4.90 | 44.6 | 0.40 | 4.00 |
| AgentCPM-GUI-8B | 21.2 | 49.5 | 41.5 | 20.4 | 13.2 | 36.4 | 56.0 | 12.6 | 14.2 | 42.9 | 1.31 | 13.7 |
| UI-TARS-DPO-7B | 9.00 | 35.6 | 56.3 | 1.74 | 7.00 | 31.8 | 60.6 | 4.04 | 14.3 | 37.5 | 1.13 | 9.43 |
| UI-TARS-1.5-7B | 25.6 | 44.9 | 60.6 | 4.04 | 4.30 | 27.2 | 64.7 | 2.55 | 23.7 | 56.6 | 0.77 | 2.06 |
| GUI-Owl-7B | 15.3 | 48.1 | 48.6 | 13.7 | 13.5 | 41.0 | 54.8 | 12.2 | 8.60 | 38.9 | 0.36 | 7.23 |

### 5.2.1 ANALYSIS OF VISUAL-GUIDED LEVEL

Given the limitations of multimodal agents in coordinate spaces, we present the results of visually-guided probing in Table 1. The results show that current multimodal agents rely heavily on tightly coupled spatial or position memory when performing GUI-based tasks, and once this alignment is perturbed, their reasoning capacity deteriorates sharply, leading to unstable behavior. In memory probing, agents fail to account for visual anomalies and instead resort to mechanical clicking, resulting in persistently low $\Delta P_{\text{Type}}$ and RS, suggesting that agents are ill-equipped to engage in reflective reasoning under anomalous conditions. Furthermore, $\Delta P_{\text{SR}}$ and VMC expose a strong bias toward selecting coordinates near the original predictions, highlighting the dependence on position memory. In reasoning probing, agents struggle to localize targets within the local context, resulting in elevated $\Delta P_{\text{SR}}$. In particular, OS-Genesis achieves 85% VMC, underscoring its heavy dependence on coordinate memorizing. Compared to SFT, RL-based and CoT-based agents (e.g. AgentCPM-GUI and GUI-Owl) mitigate memory bias and exhibit stronger reflective capability. However, this reflexivity introduces notable side effects for reasoning. Once spatial memory is disrupted, the agents also exhibit higher RS values, indicating increased over-reflection.

**Ablation and Sensitivity Analysis.** To gain a comprehensive understanding of the inference mechanisms of multimodal GUI agents, we conduct analyses across model scaling, modality ablation, non-target perturbations, parameter sensitivity, and closed-source probing (Appendix B.2.1–B.2.5). Our results show that: (i) larger models possess stronger visual memory but no better visual reasoning; (ii) visual modality is essential for grounding, with atomic instructions amplifying memory in SFT models and partially mitigating it in RL models, while their absence still induces memory behavior; (iii) perturbing non-object regions barely affects prediction, reflecting agents' dominant reliance on memorized coordinates. Moreover, stronger perturbations, focused views, or altered VMC thresholds merely rescale memory intensity. Furthermore, closed-source multimodal models remain similarly unreliable for GUI tasks, exhibiting weak anomaly detection and unstable local inference.

**Visualization and Potential Alleviation Method.** Attention visualization provides a more intuitive illustration of the memory behavior and reasoning mechanisms within multimodal GUI agents (Appendix B.2.6). Reflective prompting can reduce visual memory behavior in agents (Appendix B.2.7).

### 5.2.2 ANALYSIS OF TEXT-GUIDED LEVEL

Given the limitations of multimodal agents in the vocabulary space, we report text-guided probing results for input-dependent actions (e.g., OPENAPP, TYPE) in Table 2. At the token level, the masking of verb words (e.g., type and input) does not affect the accuracy of action-type prediction, but it does reduce the accuracy of content predictions within vocabulary space. In other words, agents cannot reliably infer input content from instructions lacking verb words. This is attributed to memory shortcuts between verbs and predictive content. Through an ablation study of masking verbs and objects, we observe that the reasoning of GUI agents is primarily driven by verbs, where removing verbs causes a far greater decline in reasoning ability than masking objects (Appendix B.3.2).

This further supports that agents rely on verb object memory shortcuts when they perform textual reasoning. At the sentence level, SFT models (e.g., OS-Genesis-7B) and most of RL-based agents suffer from training and data distribution biases. This causes instructions stored in their memory to be prioritized far above modal fusion reasoning, establishing a fixed mapping relationship between instructions and predictions. Thus, they become mere executors of misleading instructions. Notably, lower $\Delta P_{\text{SR}}$ changes (e.g., OS-ATLAS and UI-TARS-1.5) may reflect memory on fixed state (e.g., keyboards). Similarly, closed-source agents also rely on complete atomic instructions and frequently over-execution with misleading instructions (Appendix B.3.1).

Table 2: Text-guided probing of multimodal agents with token-level and sentence-level evaluation, where arrow directions represent reasoning ability.

| GUI Agents | Token-level | | Sentence-level | |
|---|---|---|---|---|
| | $\Delta P_{\text{Type}} \downarrow$ | $\Delta P_{\text{SR}} \downarrow$ | $\Delta P_{\text{Type}} \downarrow$ | $\Delta P_{\text{SR}} \downarrow$ |
| **Supervised Fine-Tuning** | | | | |
| OS-ATLAS-Pro-7B | 3.90 | 14.8 | 9.50 | 20.1 |
| OS-Genesis-7B | 30.5 | 57.1 | 67.7 | 85.4 |
| Aguvis-7B | 0.30 | 5.20 | 3.10 | 6.60 |
| UI-TARS-7B-SFT | 3.40 | 34.8 | 70.4 | 76.0 |
| **Reinforcement Learning** | | | | |
| GUI-R1-7B | 13.9 | 53.2 | 20.9 | 33.2 |
| AgentCPM-GUI-8B | 1.90 | 40.8 | 70.4 | 76.1 |
| UI-TARS-DPO-7B | 4.70 | 18.8 | 50.5 | 63.3 |
| UI-TARS-1.5-7B | 5.40 | 34.4 | 19.7 | 41.6 |
| GUI-Owl-7B | 4.97 | 49.9 | 70.1 | 97.7 |

Table 3: Structure-guided probing of multimodal agents on visual and action shortcuts. Higher values indicate stronger reliance; green = action, red = visual. Values represent 1-$\Delta P_{\text{SR}}$.

| GUI Agents | Visual Shortcuts | | | | Action Shortcuts | | | |
|---|---|---|---|---|---|---|---|---|
| | SCROLL | WAIT | PRESS | COMPLETE | SCROLL | WAIT | PRESS | COMPLETE |
| **Supervised Fine-Tuning** | | | | | | | | |
| OS-ATLAS-Pro-7B | 71.3 | 67.2 | 90.5 | 64.1 | 67.1 | 82.4 | 51.4 | 46.5 |
| OS-Genesis-7B | 19.0 | 46.9 | 27.1 | 10.9 | 60.6 | 0.00 | 84.5 | 33.6 |
| Aguvis-7B | 49.8 | 87.5 | 99.6 | 89.2 | 47.8 | 96.8 | 94.8 | 0.13 |
| UI-TARS-7B-SFT | 42.2 | 68.1 | 11.9 | 49.7 | 37.5 | 99.2 | 85.2 | 98.5 |
| **Reinforcement Learning** | | | | | | | | |
| GUI-R1-7B | 28.9 | 69.2 | 97.8 | 88.0 | 71.8 | 6.41 | 80.2 | 96.9 |
| AgentCPM-GUI-8B | 31.3 | 82.6 | 5.43 | 64.8 | 58.0 | 99.3 | 95.2 | 94.8 |
| UI-TARS-DPO-7B | 30.5 | 75.3 | 19.3 | 43.1 | 42.6 | 99.1 | 80.5 | 97.9 |
| UI-TARS-1.5-7B | 48.0 | 72.2 | 27.1 | 73.2 | 80.0 | 97.8 | 85.9 | 98.9 |
| GUI-Owl-7B | 16.8 | 76.3 | 1.83 | 75.6 | 17.9 | 0.00 | 2.23 | 74.3 |

### 5.2.3 ANALYSIS OF STRUCTURAL-GUIDED LEVEL

As discussed in Section 3, optimization for multimodal agents tends to emphasize coordinate and semantic aspects, leading to suboptimal performance on reflective and status actions. Table 3 quantifies the memory shortcuts induced by these actions in pursuit of training objectives. We observe that current models exhibit pronounced action shortcuts in WAIT and PRESS actions, which require minimal visual involvement. Early SFT models such as OS-ATLAS and Aguvis show a stronger reliance on visual shortcuts, while state-of-the-art UI-TARS and RL models further amplify action shortcuts in the COMPLETE action. Despite high accuracy from visual shortcuts, the visual modality remains largely dispensable, as action shortcuts can nearly solve the task on their own. Moreover, SCROLL results suggest that models exhibit reasoning through joint visual–semantic decision-making. Finally, GUI-Owl reflects a shift toward multimodal decision–reasoning, attributed to redesigned CoT and RL strategies. In the Appendix B.4.1, closed-source models rely on action shortcuts yet preserve strong visual reasoning, whereas open-source agents exhibit polarized shortcut use and weaker visual robustness. The case study is shown in the Appendix B.4.2.

## 6 CONCLUSION

We present Agent-ScanKit, a systematic probing toolkit for dissecting the memory and reasoning mechanisms of multimodal agents in GUI tasks. This framework reveals two core challenges: the infinite predictive space and finite generalization. Probing through three orthogonal paradigms further shows that these limitations arise from memory-dominated reasoning. These results points a crucial need for more principled RL and CoT methods to achieve robust, deployable GUI agents.

ETHICAL CONSIDERATIONS

All authors of this work have read and agree to abide by the ICLR Code of Ethics. This work systematically investigates the causes of multimodal agent unreliability and their underlying reasoning mechanisms. All experiments were conducted in controlled environments using publicly available datasets and MLLMs. The results incorporated from prior work are licensed for standard research purposes and align with their intended use. In addition, we only used LLMs solely to aid with text polishing and language refinement. No LLM-generated content contributed to the conceptual development of this paper. Our three probing strategies are mutually orthogonal, each designed to analyze whether different action types are dominated by memory or by inference. Overall, this research is centered on advancing scientific understanding of multimodal agent robustness, with the aim of encouraging the community to develop more reliable decision-making mechanisms for multimodal agents.

REPRODUCIBILITY STATEMENT

We commit that all reported results are fully reproducible in this paper. The main text specifies our experimental setup (Section 5.1), with additional details provided in the Appendix A. During the review stage, we provide supplementary materials including environment configurations, model download links, dataset preprocessing procedures, evaluation code for multimodal agents, and our sensitivity probing code. We also include sample evaluation logs to verify the authenticity of our results. We promise to release the complete codebase and preprocessing scripts to support transparency and community use.

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

# A DETAILED EXPERIMENTAL SETUP

## A.1 ACTION SPACE MAPPING

The action space $\mathcal{A}$ is parameterized to capture common user interactions in GUI environments. We define $\mathcal{A}$ as a finite set of structured actions:

$$\mathcal{A} = \big\{ \text{CLICK}(x, y), \ \text{SCROLL}(d), \ \text{TYPE}(t), \ \text{PRESSBACK}, \ \text{PRESSHOME}, \ \text{ENTER}, \tag{7}$$
$$\text{COMPLETE}, \ \text{OPENAPP}, \ \text{WAIT} \big\},$$

where

- CLICK$(x, y)$ represents a click operation at normalized coordinates $(x, y) \in [0, 1000]$ on the screen.

- SCROLL$(d)$ denotes a scroll action with discrete direction $d \in \{\text{up}, \text{down}, \text{right}, \text{left}\}$.

- TYPE$(t)$ inputs a text string $t \in \mathcal{V}^*$, where $\mathcal{V}$ is the vocabulary.

- PRESSBACK is to press the system *back* button, typically used to return to the previous screen.

- PRESSHOME is to press the system *home* button, which minimizes the current application and returns to the device's home screen.

- ENTER executes the *enter* key, often confirming an input or submitting a form.

- COMPLETE indicates the successful completion of the current task, signaling the termination of the interaction.

- OPENAPP$(t)$ launches a target application $t \in \mathcal{V}^*$ specified in the task context of Android-Control benchmark.

- WAIT pauses the agent's execution for a predefined duration, useful in asynchronous or loading scenarios.

This parameterization captures both spatially grounded actions (e.g., CLICKS) and semantic actions (e.g., TYPE and SCROLL), enabling multimodal agents to operate in realistic software environments. It should be noted that the action space $\mathcal{A}$ exclusively selects shared actions, thus standardizing the evaluation criteria.

## A.2 DETAILS OF MLLM-BASED GUI AGENTS

Table 4 provides a systematic overview of representative multimodal agents in GUI domain, highlighting their foundation models, training paradigms, and reasoning capabilities. We observe that most agents leverage either the Qwen-VL or InternVL families, with a few adopting MiniCPM-based backbones. Training strategies vary between continued pretraining (CPT) (Wu et al., 2024b), supervised fine-tuning (SFT) (Zhang et al., 2025e), and reinforcement learning (RL) (Tang et al., 2025), reflecting the necessity of end-to-end performance improvement. In particular, only a small subset of agents incorporate RL-based optimization (e.g., DPO and GRPO), and the observed improvements remain limited in practice. To investigate the impact of three probing methods on closed-source models, we extend our evaluation and analysis with GPT-4o, GPT-4-Sonnet, and GLM-4.5V. The CoT column captures whether the model produces explicit reasoning traces. We also provide the availability of official prompt resources in the last column.

## A.3 DETAILS OF BENCHMARKS

Table 5 provides a comprehensive overview of the benchmark datasets used in our evaluation, including the number of goals, screens, and the distribution of action types. This detailed characterization highlights the heterogeneity of interaction patterns across platforms, thereby evaluating the generalization of multimodal agents in task execution and environmental contexts.

Table 4: Overview of the evaluated multimodal GUI agents, including their foundation models and training paradigms. Here, CPT denotes continued pre-training on GUI tasks, SFT denotes supervised fine-tuning on GUI tasks, and RL denotes reinforcement learning on GUI tasks. CoT indicates whether the model provides explicit reasoning processes. ✔ denotes models that output reasoning for high-level goals, while directly predicting actions for low-level goals. The final column reports the availability of official prompt resources.

| GUI Agents | Foundation Model | CPT | SFT | RL | CoT | Prompt Links |
|---|---|---|---|---|---|---|
| GPT-4o | GPT-4o | ✗ | ✗ | ✗ | ✔ | https://github.com/ MadeAgents/Quick-on-the-Uptake/ blob/main/test_loop_API.py |
| Claude-4-Sonnet | Claude-4-Sonnet | ✗ | ✗ | ✗ | ✔ | |
| GLM-4.5V | GLM-4.5V | ✔ | ✔ | ✔ | ✔ | https://github.com/zai-org /GLM-V/blob/main/examples/ gui-agent/glm-45v/gui_agent_45v.py |
| OS-Atlas-Pro-4B | InternVL-2-4B | ✔ | ✔ | ✗ | ✗ | https://huggingface.co/ OS-Copilot/ |
| OS-Atlas-Pro-7B | Qwen2-VL-7B | ✔ | ✔ | ✗ | ✗ | |
| OS-Genesis-4B | InternVL-2-4B | ✗ | ✔ | ✗ | ✗ | |
| OS-Genesis-7B | Qwen2-VL-7B | ✗ | ✔ | ✗ | ✗ | |
| OS-Genesis-8B | InternVL-2-8B | ✗ | ✔ | ✗ | ✗ | |
| Aguvis-7B | Qwen2-VL-7B | ✔ | ✔ | ✗ | ✔ | https://github.com/ xlang-ai/aguvis |
| OdysseyAgent-7B | Qwen-VL-7B | ✗ | ✔ | ✗ | ✗ | https://github.com/ OpenGVLab/GUI-Odyssey/ |
| UI-TARS-2B-SFT | Qwen2-VL-2B | ✔ | ✔ | ✗ | ✔ | https://github.com/ bytedance/UI-TARS/blob/main/ codes/ui_tars/prompt.py |
| UI-TARS-7B-SFT | Qwen2-VL-7B | ✔ | ✔ | ✗ | ✔ | |
| UI-TARS-72B-SFT | Qwen2-VL-72B | ✔ | ✔ | ✗ | ✔ | |
| UI-TARS-1.5-7B | Qwen2.5-VL-7B | ✔ | ✔ | ✔ | ✔ | |
| UI-TARS-7B-DPO | Qwen2-VL-7B | ✔ | ✔ | ✔ | ✔ | |
| UI-TARS-72B-DPO | Qwen2-VL-72B | ✔ | ✔ | ✔ | ✔ | |
| GUI-R1-3B | Qwen2.5-VL-3B | ✗ | ✗ | ✔ | ✔ | https://github.com/ ritzz-ai/GUI-R1 |
| GUI-R1-7B | Qwen2.5-VL-7B | ✗ | ✗ | ✔ | ✔ | |
| AgentCPM-GUI-8B | MiniCPM-V-8B | ✔ | ✔ | ✔ | ✔ | https://huggingface.co/ openbmb/AgentCPM-GUI |
| GUI-Owl-7B | Qwen2.5-VL-7B | ✔ | ✔ | ✔ | ✔ | https://github.com/ X-PLUG/MobileAgent/tree /main/Mobile-Agent-v3/cookbook |
| GUI-Owl-32B | Qwen2.5-VL-32B | ✔ | ✔ | ✔ | ✔ | |

Table 5: Dataset statistics, including the number of goals, screens, and distribution over action types. "–" denotes that a dataset does not support a particular action type. Additionally, "Single" indicates datasets constructed for single-frame evaluation, while "Multi" refers to trajectory-level benchmarks that capture longer-horizon interactions.

| Dataset | Type | Goal | Screen | Action Space | | | | | | | |
|---|---|---|---|---|---|---|---|---|---|---|---|
| | | | | CLICK | SCROLL | TYPE | PRESS | OPENAPP | WAIT | ENTER | COMPLETE |
| AndroidControl | Multi | 1,543 | 9,987 | 5,083 | 1,211 | 632 | 343 | 608 | 1,175 | - | 1543 |
| AITZ | Multi | 506 | 4,724 | 2,736 | 601 | 500 | 265 | - | - | 118 | 506 |
| GUI-Odyssey | Multi | 1,666 | 25,651 | 16,747 | 2,622 | 2,666 | 2,044 | - | - | - | 1,572 |
| GUI-Act-Mobile | Multi | 230 | 2,079 | 1,281 | 260 | 216 | - | - | - | 92 | 230 |
| GUI-Act-Web | Multi | 66 | 316 | 97 | 149 | 26 | - | - | - | - | 44 |
| GUI-Act-Web | Single | - | 1,410 | 1,089 | 211 | - | - | - | - | - | 110 |
| OmniAct-Web | Single | - | 529 | 525 | - | - | - | - | - | - | - |
| OmniAct-DeskTop | Single | - | 1,491 | 1,491 | - | - | - | - | - | - | - |

### A.4 DETAILS OF IMPLEMENTATION

Following Zhang et al. (2025e), we evaluated 18 open source multimodal agents in five datasets using a unified benchmarking framework. Within Agent-ScanKit, for visual-guided probing, unless otherwise specified, we masked targets with a 50 black-pixel block, applied a 50-pixel edit during object modification, and in zoom-in tasks divided the screen into quadrants before selecting the target quadrant and magnifying it back to the original scale. For text-guided probing, token-level tasks replaced the initial word with `[ ]`, while sentence-level tasks injected the erroneous atomic instruction "Click the Amazon APP". For structure-guided probing, we independently corrupted the visual and textual modalities to identify the origin of memory shortcuts.

### A.5 DETAILS OF EVALUATION METRICS

For the standard metrics, Type denotes the exact match between the predicted and ground-truth action types (e.g., CLICK and SCROLL). Grounding evaluates the accuracy of GUI grounding in downstream tasks. SR measures the step-level success rate, where a step is considered successful only if both the predicted action and its associated arguments (e.g., coordinates for a click action) are correct. In our evaluation, we report the SR for CLICK, TYPE, OPENAPP, and SCROLL. For SCROLL, the direction argument (i.e., UP, DOWN, LEFT, and RIGHT) must exactly match the ground truth. For TYPE and OPENAPP, the predicted text and the ground truth must match exactly. For CLICK, following Zhang & Zhang (2024), we normalize predicted and ground-truth coordinates to 1000 and measure their relative distance. The prediction is considered correct if this distance is within 14%. For other actions (e.g., PRESSBACK), the prediction is considered correct only if it exactly matches the ground truth.

For metrics of visual-guided probing, VMC is calculated as:

$$\text{VMC} = \frac{1}{N} \sum_{i=1}^{N} \mathbb{I} \left( \left\| \mathbf{p}_i^C - \mathbf{p}_i^O \right\|_2 \leq \gamma \right), \tag{8}$$

where $\mathbf{p}_i^C \in \mathbb{R}^2$ and $\mathbf{p}_i^O \in \mathbb{R}^2$ denote the predicted coordinates in the original and masking/editing, respectively, for the $i$-th sample, and $\gamma$ is a distance threshold. $\mathbb{I}(\cdot)$ is the indicator function that returns 1 if the condition is true and 0 otherwise.

## B MORE RESULTS

In this section, we present experimental results and analysis concerning fine grained benchmark evaluations (Section B.1) and three probing methods, including ablation study, sensitivity analysis, and visualization (Section B.2 to Section B.4).

### B.1 BENCHMARK EVALUATION

#### B.1.1 DETAILED PERFORMANCE OF MULTIMODAL GUI AGENTS ACROSS ACTION TYPES

As shown in Figure 5, model performance is heavily focused on actions such as CLICK and TYPE in low- and high-level settings, while other action types exhibit varying degrees of instability. Introducing atomic instructions provides stronger textual guidance in both settings, improving accuracy. Similarly, RL-based models do not show a clear advantage over SFT counterparts. Finally, reasoning augmentation proves particularly helpful for high-level instructions, enabling models to reduce their reliance on explicit textual guidance.

#### B.1.2 DETAILED PERFORMANCE OF MULTIMODAL GUI AGENTS ACROSS TASKS AND PLATFORMS

Due to space constraints, Section 3.2 reports results only for 10 multi agents of 7∼8B scale under the AndroidControl benchmark. For completeness, Tables 6 and 7 present the accuracy of 19 GUI agents ranging from 2B to 72B across four evaluation metrics. The results further corroborate the trends discussed in the main text. Early SFT models perform poorly across almost all benchmarks, highlighting the limitations of imitation-only training. In contrast, later SFT models benefit substantially

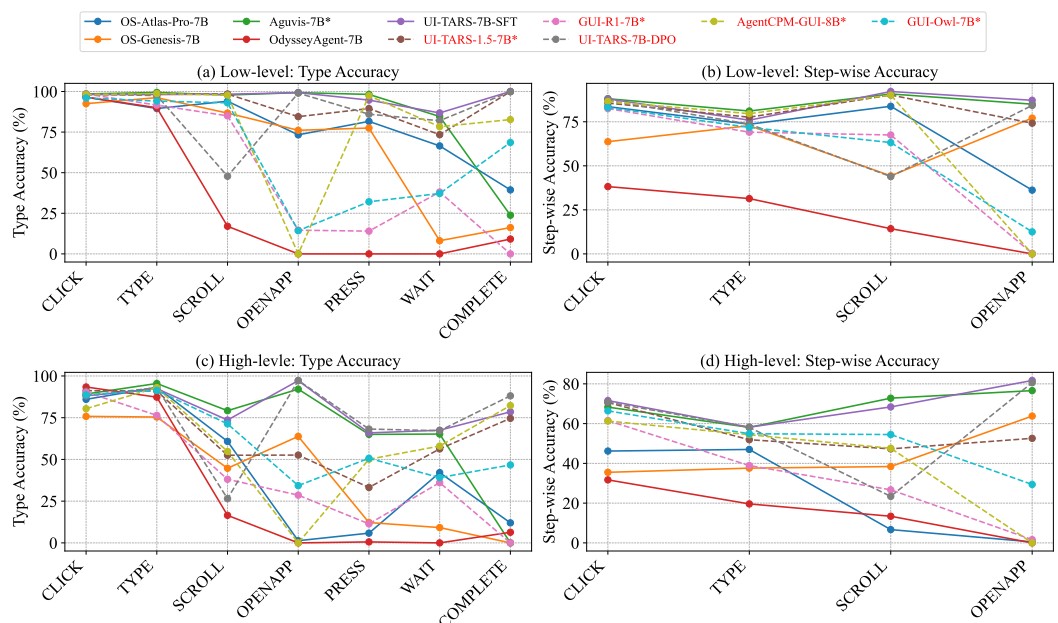

Figure 5: Detailed results of Type and SR between actions on AndroidControl Benchmark.

Table 6: Step-level and episode-level prediction performance on three GUI agent benchmarks, each containing both high-level and low-level instructions, is reported in terms of the success rates of Action Type, Grounding (Gr.), Step-wise Success Rate (SR), and Task Success. **Bold** and underlined values denote the best and second-best results, respectively.

| GUI Agents | AndroidControl-High/Low | | | | AITZ-High/Low | | | | GUI-Odyssey | | | |
|---|---|---|---|---|---|---|---|---|---|---|---|---|
| | Type | Gr. | SR | TSR | Type | Gr. | SR | TSR | Type | Gr. | SR | TSR |
| **Supervised Fine-Tuning** | | | | | | | | | | | | |
| OS-Atlas-Pro-4B | 53.3/54.0 | 27.4/27.6 | 23.9/24.6 | 0/0 | 54.6/38.8 | 24.7/13.2 | 20.7/16.9 | 0/0 | 72.6/72.5 | 31.4/30.5 | 23.6/34.1 | 0/0 |
| OS-Atlas-Pro-7B | 70.6/86.2 | 61.2/83.6 | 45.4/77.2 | 2/3 | 71.9/77.8 | 60.3/70.1 | 51.4/63.7 | 1/6 | 90.1/90.7 | 50.6/55.5 | 58.6/63.5 | 0/1 |
| OS-Genesis-4B | 42.6/69.9 | 24.4/61.1 | 16.7/45.0 | 0/0 | 30.4/65.2 | 16.3/46.9 | 11.1/45.4 | 0/0 | 26.2/46.6 | 0.53/1.06 | 5.49/9.17 | 0/0 |
| OS-Genesis-7B | 53.9/74.0 | 39.3/68.9 | 27.7/55.4 | 0/4 | 42.4/75.8 | 31.5/55.7 | 21.8/53.9 | 0/2 | 24.0/53.8 | 0.64/8.38 | 3.19/19.1 | 0/0 |
| OS-Genesis-8B | 47.8/69.4 | 27.5/54.0 | 22.7/44.6 | 0/1 | 23.6/59.3 | 12.9/37.4 | 9.71/38.1 | 0/0 | 20.0/55.4 | 0.43/2.07 | 4.02/13.0 | 0/0 |
| Aguvis-7B | 72.6/86.2 | 68.3/88.2 | 58.8/78.0 | 0/16 | 65.4/88.7 | 53.4/80.2 | 44.7/76.0 | 0/2 | 81.1/81.2 | 55.5/55.7 | 59.8/59.9 | 0/0 |
| OdysseyAgent | 56.0/58.3 | 31.7/38.2 | 20.0/24.6 | 0/0 | 53.7/61.1 | 34.4/43.6 | 25.5/31.5 | 0/0 | 79.3/77.8 | 71.3/28.4 | 34.4/33.0 | 0/0 |
| UI-TARS-2B-SFT | 81.5/97.4 | 66.7/86.1 | 67.7/87.6 | 17/49 | 76.5/98.9 | 61.5/85.1 | 58.3/86.3 | 3/26 | 71.6/83.7 | 51.1/56.9 | 45.3/55.0 | 0/0 |
| UI-TARS-7B-SFT | 83.9/**97.7** | 71.9/87.8 | 71.8/**89.6** | 22/**55** | 76.5/99.0 | 60.7/85.0 | 57.7/86.7 | 2/28 | 73.3/85.6 | 51.4/56.8 | 50.7/65.3 | 0/1 |
| UI-TARS-72B-SFT | **85.3**/97.5 | 74.6/**88.5** | **73.7**/89.6 | 23/**55** | 79.3/**99.7** | 71.1/87.9 | 63.7/**88.8** | **6**/35 | 78.6/86.6 | 56.8/58.1 | 56.7/66.4 | 0/1 |
| **Reinforcement Learning** | | | | | | | | | | | | |
| GUI-R1-3B | 60.0/77.0 | 48.5/73.7 | 38.6/62.3 | 2/9 | 53.5/79.0 | 37.9/74.1 | 26.5/56.8 | 0/0 | 67.6/86.4 | 40.4/61.6 | 35.0/62.3 | 0/1 |
| GUI-R1-7B | 64.1/75.0 | 61.5/82.5 | 44.4/62.7 | 3/5 | 55.9/84.1 | 41.2/78.5 | 28.4/57.2 | 0/0 | 73.1/91.1 | 43.8/66.6 | 37.1/61.7 | 0/2 |
| AgentCPM-GUI-8B | 75.8/94.3 | 61.3/86.5 | 61.9/85.8 | 18/47 | **85.1**/95.5 | **74.6**/83.6 | **72.3**/86.2 | **16**/32 | **92.6**/**91.4** | 62.7/60.2 | 67.8/64.3 | 1/2 |
| UI-TARS-7B-DPO | 79.7/91.1 | 70.8/87.2 | 67.2/82.6 | 22/45 | 77.5/97.4 | 65.7/85.6 | 57.4/86.7 | 2/32 | 71.9/86.5 | 53.9/61.0 | 49.7/61.6 | 0/1 |
| UI-TARS-72B-DPO | 84.0/94.2 | **75.5**/88.4 | 72.1/86.6 | **24**/49 | 78.2/96.8 | 74.3/**88.2** | 61.9/86.0 | 5/26 | 76.5/84.3 | 58.2/61.2 | 52.6/60.5 | 0/1 |
| UI-TARS-1.5-7B | 78.2/96.0 | 70.6/87.5 | 64.1/87.6 | 15/50 | 76.4/88.1 | 66.2/85.6 | 56.5/77.6 | 3/18 | 78.8/88.3 | 58.1/64.7 | 51.3/64.5 | 0/1 |
| GUI-Owl-7B | 72.8/80.7 | 66.4/83.1 | 56.9/69.0 | 9/17 | 76.7/85.1 | 59.5/69.3 | 56.7/70.0 | 2/7 | 81.4/84.9 | 68.1/**75.9** | 61.9/**70.7** | 2/**6** |
| GUI-Owl-32B | 75.0/81.4 | 71.2/86.2 | 60.4/71.5 | 10/20 | 74.3/85.6 | 55.2/71.1 | 55.4/72.7 | 3/11 | 84.4/81.5 | **73.6**/75.0 | **68.9**/69.7 | **5**/4 |

from enhanced training strategies, larger datasets, and increased model scales, achieving consistent gains and in many cases outperforming RL-based agents.

A key factor driving this improvement is the reliance on atomic instructions (Low-level), which provide strong text-level guidance and significantly boost performance. This finding suggests that current multimodal agents behave more like single-step instruction followers than genuine reasoners. Notably, AgentCPM-GUI-8B, as a representative RL-based reasoning model, demonstrates clear advantages in high-level scenarios, validating the utility of CoT reasoning. However, even in this case, performance lags behind low-level settings that supply explicit textual guidance.

Table 7: Step-level and episode-level prediction performance across GUI Agent platforms, is reported in terms of the success rates of Action Type, Grounding (Gr.), Step-wise Success Rate (SR), and Task Success. **Bold** and underlined values denote the best and second-best results, respectively.

| GUI Agents | GUIAct-Mobile | | | | GUIAct-Web-Single/Multi | | | | Omniact-Desktop | | | Omniact-Web | | |
|---|---|---|---|---|---|---|---|---|---|---|---|---|---|---|
| | Type | Gr. | SR | TSR | Type | Gr. | SR | TSR | Type | Gr. | SR | Type | Gr. | SR |
| **Supervised Fine-Tuning** | | | | | | | | | | | | | | |
| OS-Atlas-Pro-4B | 51.7 | 23.3 | 19.5 | 0 | 51.4/45.2 | 6.97/13.4 | 9.50/14.2 | -/0 | 73.2 | 15.7 | 15.6 | 39.5 | 0.95 | 0.94 |
| OS-Atlas-Pro-7B | 61.7 | 42.2 | 35.6 | 0 | 88.9/53.8 | 81.2/16.5 | 75.0/27.5 | -/3 | 99.2 | 80.6 | 80.4 | 95.8 | 79.8 | 79.2 |
| OS-Genesis-4B | 16.2 | 1.40 | 2.16 | 0 | 78.4/31.0 | 58.5/13.4 | 54.8/9.17 | -/0 | 0.48 | 0.00 | 0.00 | 0.56 | 0.00 | 0.00 |
| OS-Genesis-7B | 24.9 | 3.35 | 5.77 | 0 | 84.8/32.6 | 70.6/13.4 | 64.7/13.0 | -/3 | 73.9 | 12.8 | 12.4 | 79.2 | 25.5 | 25.3 |
| OS-Genesis-8B | 11.9 | 1.32 | 1.53 | 0 | 65.1/32.9 | 46.4/10.3 | 36.2/14.2 | -/0 | 0.69 | 0.00 | 0.00 | 0.37 | 0.00 | 0.00 |
| Aguvis-7B | 50.5 | 33.0 | 28.6 | 0 | 82.6/50.6 | 81.1/46.4 | 71.2/40.8 | -/15 | 90.6 | 73.8 | 71.5 | 93.0 | 78.6 | 78.1 |
| OdysseyAgent | 57.4 | 18.0 | 11.7 | 0 | 75.5/29.4 | 3.94/2.06 | 3.04/1.26 | -/0 | 95.1 | 26.1 | 26.0 | 90.9 | 26.8 | 26.6 |
| UI-TARS-2B-SFT | 72.9 | 52.0 | 42.8 | 0 | 81.7/57.3 | 64.4/38.1 | 61.6/47.5 | -/6 | 93.6 | 61.4 | 59.4 | 93.5 | 67.8 | 67.2 |
| UI-TARS-7B-SFT | 19.2 | 15.7 | 12.2 | 0 | 86.8/45.2 | 73.9/44.3 | 70.6/33.2 | -/3 | 90.8 | 63.3 | 61.4 | 93.3 | 73.9 | 73.3 |
| UI-TARS-72B-SFT | 64.9 | 52.1 | 46.0 | 1 | 89.8/58.5 | 72.4/54.6 | 69.4/49.4 | -/6 | 96.2 | 77.0 | 74.8 | 99.4 | 78.1 | 78.1 |
| **Reinforcement Learning** | | | | | | | | | | | | | | |
| GUI-R1-3B | 48.4 | 16.3 | 21.8 | 0 | 43.4/27.2 | 5.97/9.27 | 7.16/10.5 | -/3 | 86.0 | 77.6 | 68.2 | 96.2 | 75.2 | 74.7 |
| GUI-R1-7B | 57.9 | 27.3 | 20.9 | 0 | 69.4/35.4 | 12.0/17.5 | 10.1/13.6 | -/2 | 85.0 | 79.6 | 69.9 | 96.6 | 81.0 | 80.6 |
| AgentCPM-GUI-8B | **74.7** | **61.0** | **58.2** | **8** | 72.5/44.3 | 40.9/14.4 | 44.6/29.7 | -/3 | 68.3 | 44.7 | 44.3 | 75.3 | 46.8 | 43.9 |
| UI-TARS-7B-DPO | 53.6 | 45.6 | 36.7 | 0 | 87.6/50.0 | 74.4/60.8 | 70.7/39.9 | -/3 | 89.4 | 64.1 | 61.9 | 96.9 | 70.6 | 70.1 |
| UI-TARS-72B-DPO | 67.3 | 53.3 | 46.6 | 2 | 86.9/38.3 | 73.0/56.7 | 67.6/28.2 | -/4 | 85.4 | 75.3 | 65.9 | 99.6 | 79.6 | 79.4 |
| UI-TARS-1.5-7B | 68.6 | 41.1 | 36.7 | 0 | 87.2/57.3 | 70.6/44.3 | 67.1/42.7 | -/4 | 92.3 | 51.5 | 49.8 | 98.5 | 84.8 | 84.2 |
| GUI-Owl-7B | 62.9 | 45.3 | 41.1 | 1 | 82.1/38.6 | 62.0/36.1 | 59.8/27.2 | -/3 | 88.6 | 65.4 | 65.2 | 91.7 | 71.5 | 70.4 |
| GUI-Owl-32B | 60.9 | 40.9 | 38.3 | 1 | 87.5/47.2 | 64.7/47.4 | 69.3/28.5 | -/5 | 92.5 | 71.1 | 71.0 | 96.0 | 74.5 | 73.9 |

Despite these advances, generalization remains severely limited. Once extended to out-of-domain tasks or new environments, all models exhibit sharp performance degradation, underscoring that current GUI agents operate primarily under a memory-driven paradigm rather than robust reasoning-based generalization.

## B.2 FURTHER RESULTS OF VISUAL-GUIDED PROBING EXPERIMENTS

### B.2.1 COMPARISON OF CLOSED- AND OPEN- SOURCE MODELS

To evaluate how closed source agents behave under visual guided probing, we extend our analysis and compare them with open source models on the AITZ benchmark, as shown in Table 8. Among closed source systems, newer models such as GLM-4.5V, which integrate GUI oriented capabilities, show higher RS under memory probing and robust reasoning under Zoom in probing. In contrast, GPT-4o shows unreliable when object are masked or edited and undergoes a clear reasoning collapse under Zoom in probing. Furthermore, the overall robustness of closed source models stays broadly comparable to SFT based open source models. Within the SFT group, OS-ATLAS-Pro-7B shows desirable anomaly sensitivity, whereas RL trained agents such as AgentCPM-GUI-8B and GUI-Owl-7B are even more responsive to perturbations and tend to maintain relatively stable reasoning under Zoom-in setting. Thus, general purpose multimodal models remain unreliable when performing GUI

tasks, fail to detect visual anomalies, and are unable to guarantee reliable inference in local scenarios.

Table 8: Comparison of visual-guided probing on closed- and open-source multimodal agents in GUI tasks on the AITZ benchmark. Memory robustness is evaluated through object masking and object editing, while reasoning ability is assessed via the zoom-in setting. The masking/editing ratio and the VMC distance threshold are both fixed at 50 pixels. $\Delta P_{\text{Type}} \downarrow$ and $\Delta P_{\text{SR}} \downarrow$ denote changes in prediction performance, where the direction of the arrows indicates greater reasoning ability.

| GUI Agents | Object Masking | | | | Object Editing | | | | Zoom-in | | | |
|---|---|---|---|---|---|---|---|---|---|---|---|---|
| | $\Delta P_{\text{Type}} \uparrow$ | $\Delta P_{\text{SR}} \uparrow$ | VMC↓ | RS↑ | $\Delta P_{\text{Type}} \uparrow$ | $\Delta P_{\text{SR}} \uparrow$ | VMC↓ | RS↑ | $\Delta P_{\text{Type}} \downarrow$ | $\Delta P_{\text{SR}} \downarrow$ | VMC↓ | RS↓ |
| **Closed-source Models** | | | | | | | | | | | | |
| GPT-4o | 9.07 | 34.4 | 44.3 | 1.74 | 8.44 | 35.3 | 42.1 | 1.40 | 15.8 | 99.0 | 38.2 | 0.36 |
| GLM-4.5V | 13.3 | 37.8 | 49.5 | 6.64 | 12.5 | 41.1 | 49.7 | 8.00 | 2.69 | 71.4 | 0.05 | 1.55 |
| Claude-4-Sonnet | 14.1 | 40.4 | 40.4 | 2.17 | 1.00 | 31.4 | 53.3 | 0.99 | 1.77 | 97.2 | 14.1 | 1.21 |
| **Open-source Models** | | | | | | | | | | | | |
| OS-ATLAS-Pro-7B | 19.2 | 54.6 | 26.6 | 17.8 | 17.4 | 56.0 | 25.2 | 16.1 | 33.7 | 99.6 | 0.41 | 32.9 |
| OS-Genesis-7B | 0.99 | 2.03 | 96.1 | 0.39 | 0.73 | 2.76 | 96.4 | 0.52 | 9.13 | 99.6 | 73.4 | 5.09 |
| UI-TARS-7B-SFT | 1.40 | 28.4 | 64.8 | 0.99 | 1.88 | 33.7 | 58.5 | 1.19 | 1.32 | 52.0 | 5.04 | 0.51 |
| GUI-R1-7B | 12.0 | 45.3 | 43.1 | 11.5 | 11.6 | 45.3 | 40.8 | 10.6 | 1.63 | 61.4 | 2.37 | 1.35 |
| AgentCPM-GUI-8B | 29.6 | 51.7 | 33.2 | 22.8 | 23.2 | 41.9 | 40.3 | 22.1 | 7.38 | 46.4 | 1.52 | 7.11 |
| GUI-Owl-7B | 30.9 | 60.7 | 35.4 | 29.4 | 29.3 | 58.1 | 36.6 | 27.7 | 8.02 | 30.9 | 1.68 | 7.53 |

### B.2.2 IMPACT OF MODEL SCALING LAW

To evaluate how model scaling law influences robustness under visual-guided probing, we systematically evaluate 18 multimodal GUI agents, ranging from 2B to 72B parameters. Figure 6 presents the scaling trends of four metrics, allowing us to quantify whether larger models exhibit stronger visual reasoning or simply accumulate greater spatial memory. The detailed analyses are provided below.

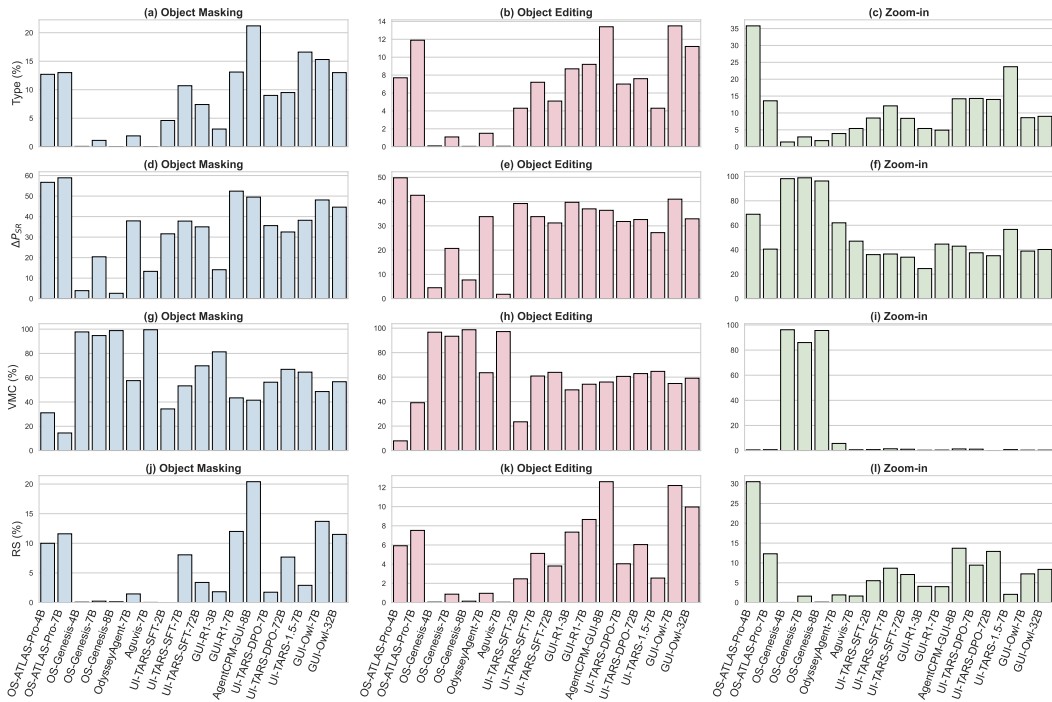

Figure 6: Comprehensive robustness profiling of 18 GUI agents across four evaluation dimensions and three visual-perturbation settings.

(i) $\Delta P_{\text{Type}}$ and RS: As we have observed, even across different parameter scales, all models maintain low $\Delta P_{\text{Type}}$ under both memory probes and the reasoning probe. This indicates that an increase in the parameter does not reduce the tendency to resort to clicking in memory-driven scenarios. In general, smaller models exhibit stronger memory-dominant behavior, as seen in the GUI-R1 and UI-TARS-SFT series. Correspondingly, their RS values are lower, suggesting impulsive action selection rather than deliberate reasoning. However, under magnified conditions, larger models exhibit higher $\Delta P_{\text{Type}}$ and RS values, indicating that they over-reflect when the global visual space is disturbed.

(ii) $\Delta P_{\text{SR}}$: In memory probing tasks, the $\Delta P_{\text{SR}}$ of SFT models generally stays below that of RL models and expose stronger visual memory behavior. An increase in parameters slightly reduces $\Delta P_{\text{SR}}$ and indicates memory expansion. In the Zoom in setting, as parameters increase, the $\Delta P_{\text{SR}}$ of most models increase to approximately 40% and show that the introduction of various strategies fails to enhance local reasoning ability, including models such as UI-TARS-SFT (2 to 72B) and GUI-Owl (7B to 32B).

(iii) VMC: it directly reflects agents maintains predictive consistency before and after perturbations if relying on visual memory. First, for both SFT and RL models, increasing the number of parameters generally yields stronger memory capacity, as illustrated by the UI-TARS-SFT, GUI-R1, and GUI-Owl series. Second, even with larger model sizes, SFT models consistently exhibit higher VMC values than their RL counterparts, indicating a stronger tendency to select positions near the target based on memorized spatial priors. Under Zoom-in perturbations, the OS-Genesis series likewise shows pronounced positional memory, whereas other models generate inferred predictions despite overall lower performance.

### B.2.3 ABLATION STUDY OF VISUAL AND TEXTUAL MODALITIES ON VISUAL-GUIDED PROBING

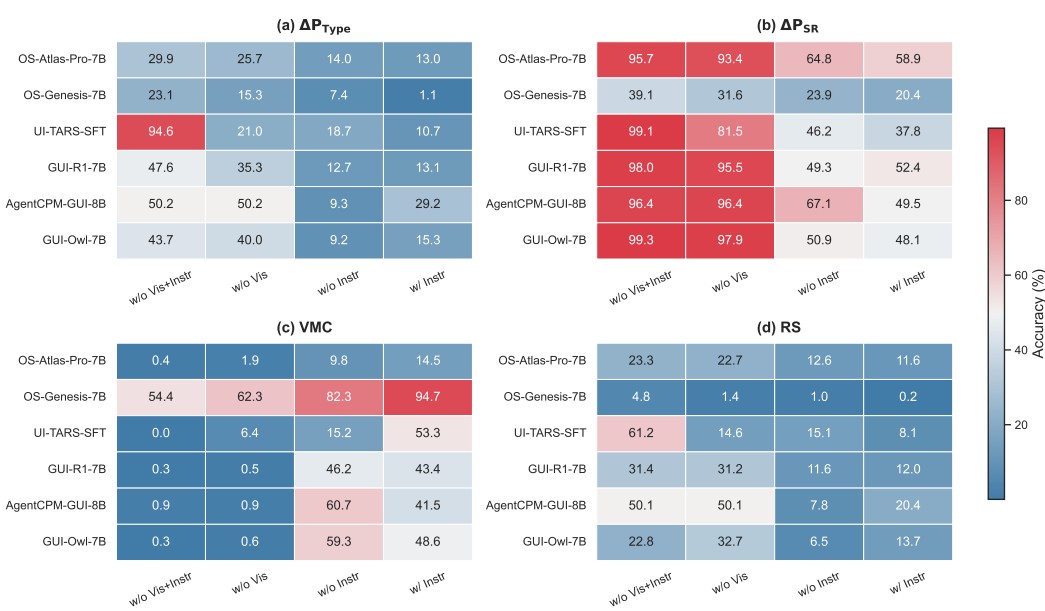

Figure 7: Ablation study of different settings in the object-masking of visual-guided probing. "Vis" = Visual Modality, "Instr" = Atomic Instruction.

To further investigate the role of visual and textual modalities in visual grounding, we conduct an ablation analysis on object-masking under four conditions, as shown in Figure 7. Specifically,

(i) $\Delta P_{\text{Type}}$ and RS: Except for UI TARS-7B-SFT, all models show failure of atomic instructions when visual modalities are absent, meaning the models rely on rote memorization of action type predictions. In general, SFT models show stronger memory retention but weaker reflection compared to RL models. When integrated with visual modalities, atomic instructions increase both $\Delta P_{\text{Type}}$ and

RS in RL models and become essential for reflective decision. In contrast, atomic instructions trigger memory activation in SFT models.

(ii) $\Delta P_{SR}$: Most agents cannot achieve visual localization without visual modality involvement. In contrast, SFT models possess stronger and larger memories; for instance, UI-TARS-7B-SFT activates an extent visual memories after the incorporation of atomic instructions. After visual modality integration, atomic instructions act as triggers that activate all model memories and reduce the model $\Delta P_{SR}$. However, even without this integration, the model still shows robust visual memory capabilities.

(ii) VMC: Consistently, without visual input, most agents are unable to recall positional memory, with the exception of OS-Genesis-7B. After the integration of the visual modality, atomic commands reduce positional memory in RL models and enhance it in SFT models. This pattern indicates that SFT models follow visual reasoning in a mindless manner, whereas RL models interpret such commands as conflicting or erroneous instructions.

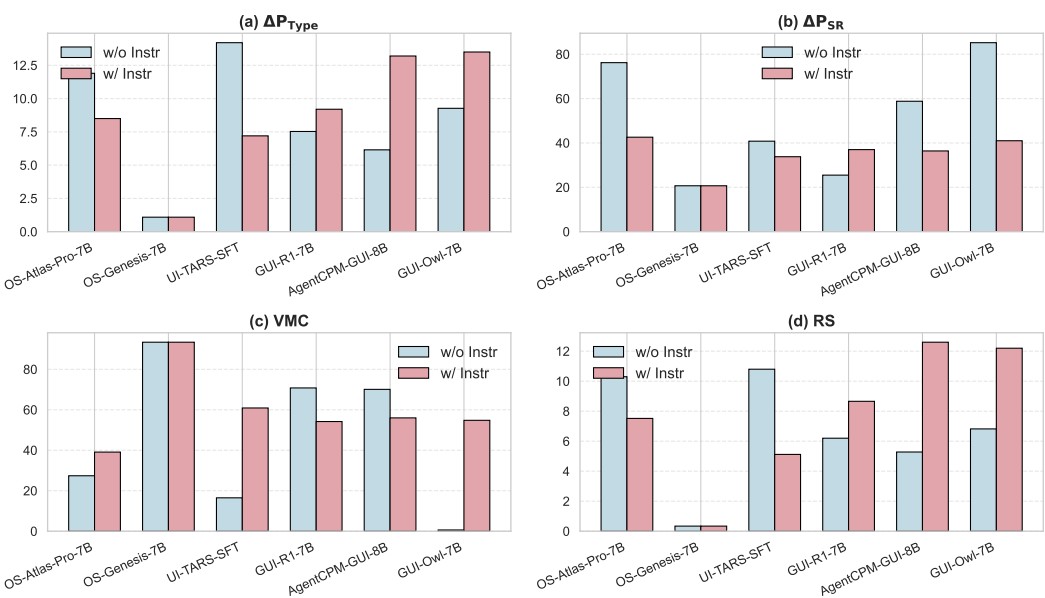

Figure 8: Ablation study of atomic instruction in the object editing of visual guided probing.

Correspondingly, as shown in Figure 8, we observe the same phenomenon in the object editing probing. Therefore, a small number of models show absolute positional memory. After the integration of visual modalities, atomic instructions serve distinct functions for SFT and RL. Nevertheless, even without atomic instructions, existing agents show unreliable reasoning.

### B.2.4    IMPACT OF NON-OBJECT MASKING AND EDITING

To evaluate whether perturbing non-object regions affects agent behavior, we introduce a control experiment that randomly masks or edits $50\times50$ patches only outside the ground truth target region. As shown in Table 11, all agents show minimal changes across all four metrics. In particular, VMC remains extremely high from 62% to 100%, indicating that models continue to click confidently on memoried coordinates, and RS stays near zero, showing almost no reflective behavior. These findings indicate that disturbances to non-object regions exert negligible influence on model decision-making: (i) When the target region remains intact, memory-driven rapid decision-making predominantly prevails; (ii) The global visual environment remains largely unchanged, thereby safeguarding the model's memory-driven decision retrieval.

### B.2.5    SENSITIVITY ANALYSIS OF VISUAL-GUIDED PROBING

**Masking/Editing Ratio.**    To investigate how the severity of object masking and editing affects the memory behavior of GUI agents, we perform a scaling analysis by gradually increasing the

Table 9: Impact of masking or editing non-object regions on visual-guided probing.

| GUI Agents | Non-object masking | | | | Non-object editing | | | |
|---|---|---|---|---|---|---|---|---|
| | $\Delta P_{\text{Type}}$ | $\Delta P_{\text{SR}}$ | VMC | RS | $\Delta P_{\text{Type}}$ | $\Delta P_{\text{SR}}$ | VMC | RS |
| **Supervised Fine-tuning** | | | | | | | | |
| OS-ATLAS-Pro-7B | 2.23 | 8.37 | 62.6 | 2.00 | 2.22 | 8.68 | 62.7 | 2.00 |
| OS-Genesis-7B | 2.32 | 22.2 | 87.7 | 0.72 | 2.39 | 22.8 | 87.4 | 0.82 |
| UI-TARS-7B-SFT | 2.29 | 4.79 | 92.5 | 1.58 | 1.88 | 4.89 | 92.0 | 0.94 |
| **Reinforcement Learning** | | | | | | | | |
| GUI-R1-7B | 1.70 | 7.56 | 93.4 | 1.47 | 2.01 | 8.21 | 92.7 | 1.74 |
| AgentCPM-GUI-8B | 1.91 | 6.19 | 93.9 | 1.77 | 1.33 | 5.86 | 94.0 | 1.21 |
| GUI-Owl-7B | 2.13 | 33.9 | 93.5 | 1.83 | 0.00 | 30.0 | 100 | 0.00 |

perturbation ratio from 10% to 100% across six multimodal GUI agents. Table 10 summarizes the resulting changes in four metrics under both perturbation types. Specifically, even under a larger masking ratio, agents overwhelmingly preserve positional memory, as reflected by consistently high VMC values. For example, object masking still ranges from 9.87% to 92.5%, and object editing is 21.6% to 93.4%. This indicates a systematic tendency to predict coordinates near memorized objects despite the larger removal of visual evidence. Moreover, an increase in the perturbation ratio has minimal effect on models with strong memory bias, particularly SFT variants, whose VMC and RS values remain largely invariant. Notably, the highest reflective action score is only 25.6% for AgentCPM-GUI-8B at a 100% masking rate. In contrast, RL and CoT models exhibit more robust reasoning than SFT models. Models such as AgentCPM-GUI-8B and GUI-Owl-7B exhibit larger changes of SR under heavy perturbations, which highlights a degree of reflection that re chooses a potential correct area. However, OS Genesis remains nearly invariant across ratios, showing absolute position memory, while OS ATLAS-Pro-7B also shows unusually high sensitivity due to its anomaly oriented training. Therefore, the ratio ablation shows that stronger perturbations do not induce stronger reflection. Instead, models may explore and locate alternative positions, which may result in less reliable autonomous behavior.

**Zoom-in ratio.** To evaluate how different visual focus mechanisms affect the reasoning capabilities exhibited by GUI agents, we provide two zoom-in detection designs: (i) quadrant cropping, retaining only 25% of the visual space containing the target; (ii) horizontal splitting, preserving a larger proportion of the global layout. Table 11 presents comparative results. Horizontal segmentation yields markedly lower $\Delta P_{\text{Type}}$, $\Delta P_{\text{SR}}$, and RS values, indicating that preserving larger portions of the visual context substantially improves reasoning stability. This confirms that current agents cannot reliably reason under focused-view conditions. Once global visual information is reduced, their decisions become unreliability. Specifically, variations in action type ($\Delta P_{\text{Type}}$) remain relatively small, largely because these predictions may driven by atomic instruction. In contrast, large fluctuations in visual positioning ($\Delta P_{\text{SR}}$) reveal that visual perception depends heavily on recalling global structural patterns. Moreover, richer context also suppresses over-reflection behavior. Notably, OS-Genesis-7B and GUI-R1-7B show the strongest degradation under quadrant-based Zoom-in, reflecting their reliance on global layout memory. Overall, these results demonstrate that reasoning under visual focus critically depends on the preservation of structural context; although quadrant partitioning retains the task's target region, it removes essential layout cues and prevents agents from performing reliable visual reasoning.

**Impact of VMC threshold.** To evaluate the impact of VMC threshold for reflecting an agent's positional-memory reliance, we vary the VMC distance threshold across object masking, object editing, and zoom-in perturbations. As shown in Figure 9, although increasing the VMC threshold amplifies the measured positional memory, this effect gradually saturates beyond 20%, indicating that VMC is a stable diagnostic of memory-driven behavior. In object masking and editing, models display absolute positional memory at the 0% threshold; between 0% and 20%, VMC rises sharply, indicating predictions remain tightly clustered near memorized coordinates, confirming reliance on

Table 10: Scaling analysis of visual-guided probing across increasing masking and editing ratios (10%~100%) for six multimodal GUI agents.

| GUI Agents | Ratio | Object Masking | | | | Object Editing | | | |
|---|---|---|---|---|---|---|---|---|---|
| | | $\Delta P_{Type}$ | $\Delta P_{SR}$ | VMC | RS | $\Delta P_{Type}$ | $\Delta P_{SR}$ | VMC | RS |
| OS-ATLAS-Pro-7B | 10 | 8.78 | 44.2 | 20.8 | 7.63 | 2.82 | 11.1 | 61.3 | 2.33 |
| | 30 | 11.2 | 51.9 | 16.9 | 9.91 | 7.34 | 35.6 | 43.7 | 6.66 |
| | 50 | 13.0 | 58.9 | 14.5 | 11.6 | 8.50 | 42.6 | 39.1 | 7.52 |
| | 70 | 14.6 | 65.3 | 12.1 | 13.1 | 11.9 | 57.3 | 26.9 | 10.9 |
| | 100 | 16.5 | 72.8 | 9.87 | 15.4 | 13.5 | 63.3 | 21.6 | 12.5 |
| OS-Genesis-7B | 10 | 1.00 | 20.0 | 95.5 | 0.21 | 1.09 | 20.7 | 93.4 | 0.34 |
| | 30 | 1.00 | 20.2 | 95.2 | 0.23 | 1.09 | 20.7 | 93.4 | 0.34 |
| | 50 | 1.11 | 20.4 | 94.7 | 0.23 | 1.09 | 20.7 | 93.4 | 0.34 |
| | 70 | 1.27 | 20.6 | 94.1 | 0.28 | 1.09 | 20.7 | 93.4 | 0.34 |
| | 100 | 1.36 | 21.2 | 92.5 | 0.32 | 1.09 | 20.7 | 93.4 | 0.34 |
| UI-TARS-7B-SFT | 10 | 8.30 | 29.3 | 64.8 | 6.02 | 1.98 | 6.10 | 91.9 | 1.00 |
| | 30 | 9.40 | 33.2 | 59.3 | 7.03 | 5.03 | 20.7 | 73.4 | 3.30 |
| | 50 | 10.7 | 37.8 | 53.3 | 8.05 | 7.20 | 33.8 | 60.9 | 5.12 |
| | 70 | 12.3 | 42.6 | 46.9 | 9.21 | 8.62 | 41.5 | 53.4 | 6.12 |
| | 100 | 14.8 | 49.1 | 39.5 | 11.2 | 10.5 | 46.7 | 47.3 | 7.53 |
| GUI-R1-7B | 10 | 9.61 | 36.6 | 61.3 | 8.94 | 1.59 | 7.35 | 93.8 | 1.34 |
| | 30 | 11.3 | 44.5 | 52.8 | 10.7 | 5.44 | 24.0 | 71.9 | 4.99 |
| | 50 | 13.1 | 52.4 | 43.4 | 12.0 | 9.20 | 37.0 | 54.2 | 8.66 |
| | 70 | 14.5 | 61.2 | 34.5 | 14.3 | 11.5 | 45.4 | 42.8 | 10.9 |
| | 100 | 18.8 | 69.5 | 25.2 | 17.8 | 14.7 | 52.4 | 32.8 | 13.9 |
| AgentCPM-GUI-8B | 10 | 14.3 | 35.8 | 57.7 | 13.6 | 1.73 | 7.06 | 92.9 | 1.55 |
| | 30 | 18.4 | 44.0 | 48.0 | 17.6 | 8.59 | 25.7 | 69.2 | 8.18 |
| | 50 | 21.2 | 49.5 | 41.5 | 20.4 | 13.2 | 36.4 | 56.0 | 12.6 |
| | 70 | 23.3 | 45.3 | 36.0 | 22.4 | 15.7 | 43.3 | 48.3 | 15.0 |
| | 100 | 26.6 | 60.9 | 28.8 | 25.6 | 18.9 | 49.7 | 42.2 | 18.2 |
| GUI-Owl-7B | 10 | 11.9 | 39.2 | 58.8 | 10.3 | 3.04 | 8.27 | 90.7 | 2.65 |
| | 30 | 13.8 | 44.4 | 52.7 | 12.2 | 9.84 | 28.0 | 68.5 | 8.78 |
| | 50 | 15.3 | 48.1 | 48.6 | 13.7 | 13.5 | 41.0 | 54.8 | 12.2 |
| | 70 | 16.5 | 52.1 | 44.4 | 14.8 | 16.1 | 48.9 | 45.5 | 14.6 |
| | 100 | 17.6 | 55.5 | 40.6 | 15.9 | 19.9 | 56.8 | 36.4 | 18.1 |

Table 11: Comparison of zoom-in probing strategies using 4-way quadrant crops and 2-way vertical splits across six multimodal GUI agents.

| GUI Agents | Zoom-in (4-way Quadrant) | | | | Zoom-in (2-way Horizontal Split) | | | |
|---|---|---|---|---|---|---|---|---|
| | $\Delta P_{Type} \downarrow$ | $\Delta P_{SR} \downarrow$ | VMC$\downarrow$ | RS$\downarrow$ | $\Delta P_{Type} \downarrow$ | $\Delta P_{SR} \downarrow$ | VMC$\downarrow$ | RS$\downarrow$ |
| **Supervised Fine-tuning** | | | | | | | | |
| OS-ATLAS-Pro-7B | 13.6 | 40.5 | 0.72 | 12.3 | 5.94 | 18.6 | 0.28 | 4.88 |
| OS-Genesis-7B | 2.90 | 98.8 | 86.0 | 1.61 | 5.79 | 68.0 | 69.4 | 1.77 |
| UI-TARS-7B-SFT | 12.1 | 36.5 | 1.43 | 8.66 | 4.10 | 11.8 | 1.49 | 1.59 |
| **Reinforcement Learning** | | | | | | | | |
| GUI-R1-7B | 4.90 | 44.6 | 0.40 | 4.00 | 2.71 | 34.5 | 0.74 | 2.11 |
| AgentCPM-GUI-8B | 14.2 | 42.9 | 1.31 | 13.7 | 4.39 | 13.9 | 3.27 | 3.73 |
| GUI-Owl-7B | 8.60 | 38.9 | 0.36 | 7.23 | 5.14 | 12.9 | 0.42 | 4.30 |

spatial memory rather than perception. RL/CoT models—particularly AgentCPM-GUI-8B and GUI-Owl-7B—show slower VMC growth, reflecting weaker positional memory, whereas SFT models demonstrate either absolute position memory or consistent clustering near the target (e.g., OS-Genesis-

7B and UI-TARS-7B-SFT). OS-ATLAS-Pro-7B is notably sensitive to perturbations, often attempting exploration. In the Zoom-in setting, due to the global spatial structure is disrupted, most agents shift to inference regardless of the VMC threshold, with OS-Genesis-7B being the only exception that retains absolute positional memory. Overall, varying the VMC threshold does not alter the memory behaviors of GUI agents, while simply rescales the magnitude of their underlying memory effects.

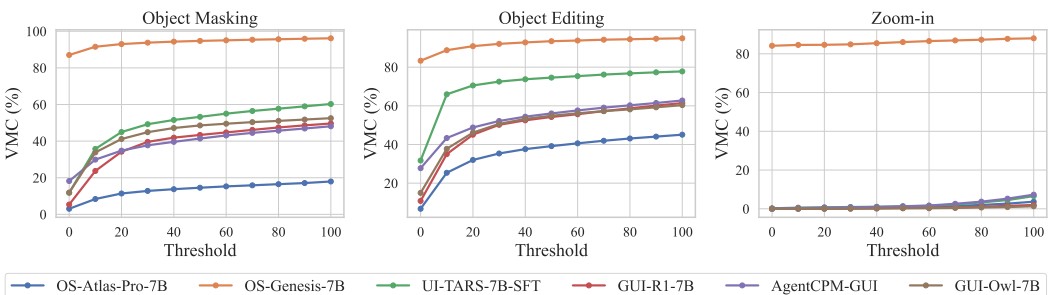

Figure 9: Impact of varying VMC distance thresholds under object masking, object editing, and zoom-in perturbations across six multimodal GUI agents.

### B.2.6 VISUALIZATION OF MULTIMODAL AGENTS ATTENTION

Following Yan & Zhang (2025) and Zhang et al. (2025c), we adopt a relative attention-based visualization method to display the attention regions of multimoda agents. As shown in Figure 10, the SFT model and UI-TARS-DPO preserve attention to object regions even under masking due to memory bias, thus generating coordinates consistent with the original. In contrast, GUI-R1 and UI-TARS-1.5 detect occlusions in the target areas, redirecting actions to the search box and the application details page, respectively. As shown in Figure 11, Aguvis and UI-TARS-DPO continue to exhibit memory-driven behavior in object editing, while OS-Atlas, UI-TARS-SFT, and UI-TARS-1.5 accomplish the task through exploratory strategies. Interestingly, GUI-Owl, equipped with CoT analysis, instinctively taps the target area under masking but switches to exploration during editing. We attribute this inconsistency to mechanical CoT generation. This enables memory recall rather than adaptive reasoning.

### B.2.7 POTENTIAL ALLEVIATION STRATEGIES

To mitigate the phenomenon of agents taking visual shortcuts, we adopt reflection regularization approaches as shown in Table 12. The reflective regularization prompt is provided below.

> **Reflective Regularization Prompt**
>
> Please note that if the object area cannot be perceived, endeavor to employ reflective actions such as SCROLL, PRESSBACK, PRESSHOME, WAIT, or COMPLETE to terminate the task.

The results indicate that these models are default to memory-driven under perturbations, but exhibit inherently uncertain. Thus, incorporating lightweight reflective regularization can relatively guide the agent's behavior away from memory-driven towards more deliberate action selection. Across SFT models, reflective regularization consistently raises both $\Delta P_{\text{Type}}$ and $\Delta P_{\text{SR}}$ while lowering VMC or improving RS. RL/CoT models exhibit more heterogeneous behaviors. For GUI-R1 and GUI-Owl, reflective cues effectively suppress over-optimized atomic policies, reducing their tendency toward impulsive shortcut execution. In contrast, AgentCPM-GUI shows only marginal gains, reflecting that the model has lost its generality and can only perform tasks based on internalized prompts.

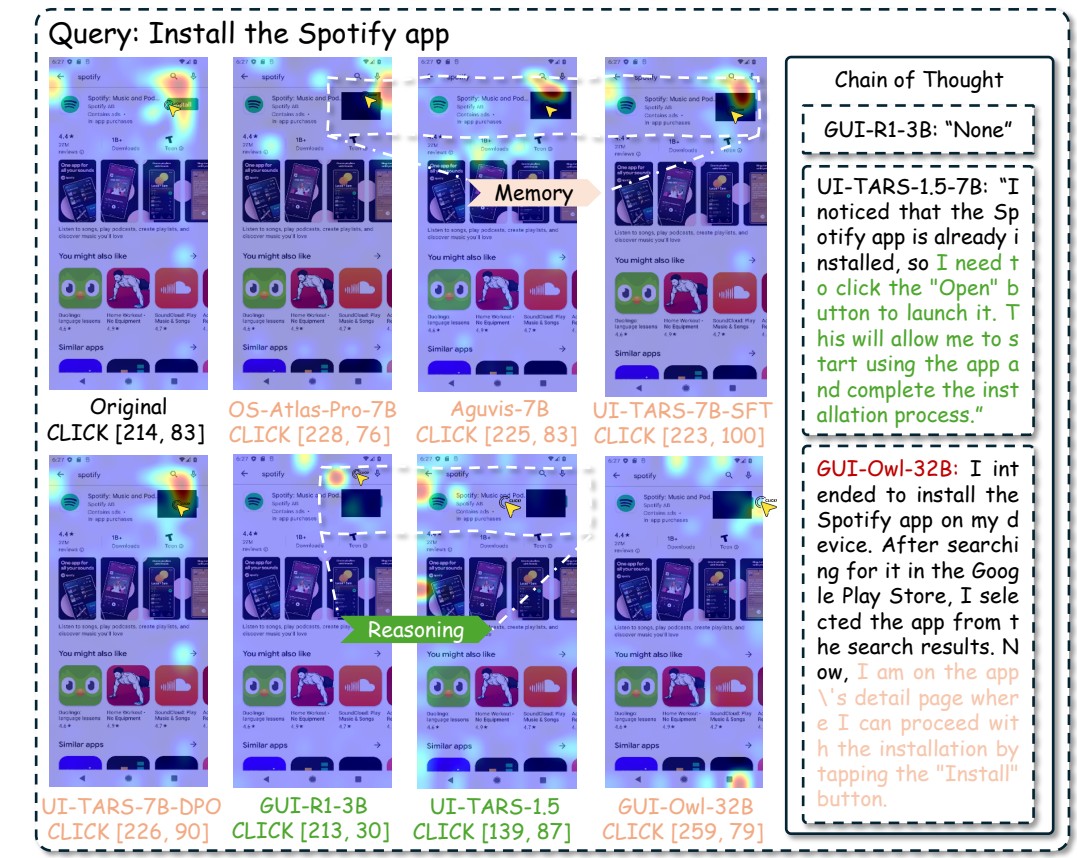

Figure 10: Visualization of relative attention in middle-layer during the decision-making process of Qwen-VL-based multimodal agents in the object masking probing. The right-side illustrate the reasoning trajectory with an integrated CoT agents.

### B.3 FURTHER RESULTS OF TEXTUAL-GUIDED PROBING EXPERIMENTS

#### B.3.1 COMPARISON OF CLOSED- AND OPEN- SOURCE MODELS

To analyze how closed-source agents behave under textual-guided probing, we extend our analysis and compare them with open-source models on the AITZ benchmark. As shown in Table 13, The results also show two dominant weaknesses: token-level brittleness and sentence-level over-compliance.

Specifically, at the token level, most models can still predict the TYPE action, but SR drops significantly, especially for GLM-4.5V, OS-Genesis-7B, and RL-based agents, indicating a strong dependence on full atomic instructions. Closed-source models (e.g., GPT-4o and Claude-4-Sonnet) and SFT variants degrade less, instead demonstrating superior contextual reasoning abilities. At the sentence level, several agents over-follow erroneous instructions despite conflicting visual evidence, including GPT-4o, Claude-4-Sonnet, and GUI-R1-7B, whereas GLM-4.5V and SFT models are better at rejecting misleading commands. Models such as AgentCPM-GUI-8B and GUI-Owl-7B predict the action type but fail to infer correct textual. Therefore, current GUI agents struggle to integrate visual grounding with instruction reasoning, defaulting to memory shortcuts either atomic instruction templates or instruction-following modes.

#### B.3.2 IMPACT OF MASKING VERB VS. OBJECT TOKENS AT TOKEN-LEVEL PROBING

To quantify which token types most strongly affect model reasoning, we conduct token-level textual probing by independently masking verbs (e.g., type and input) and random tokens in the object (i.e.,

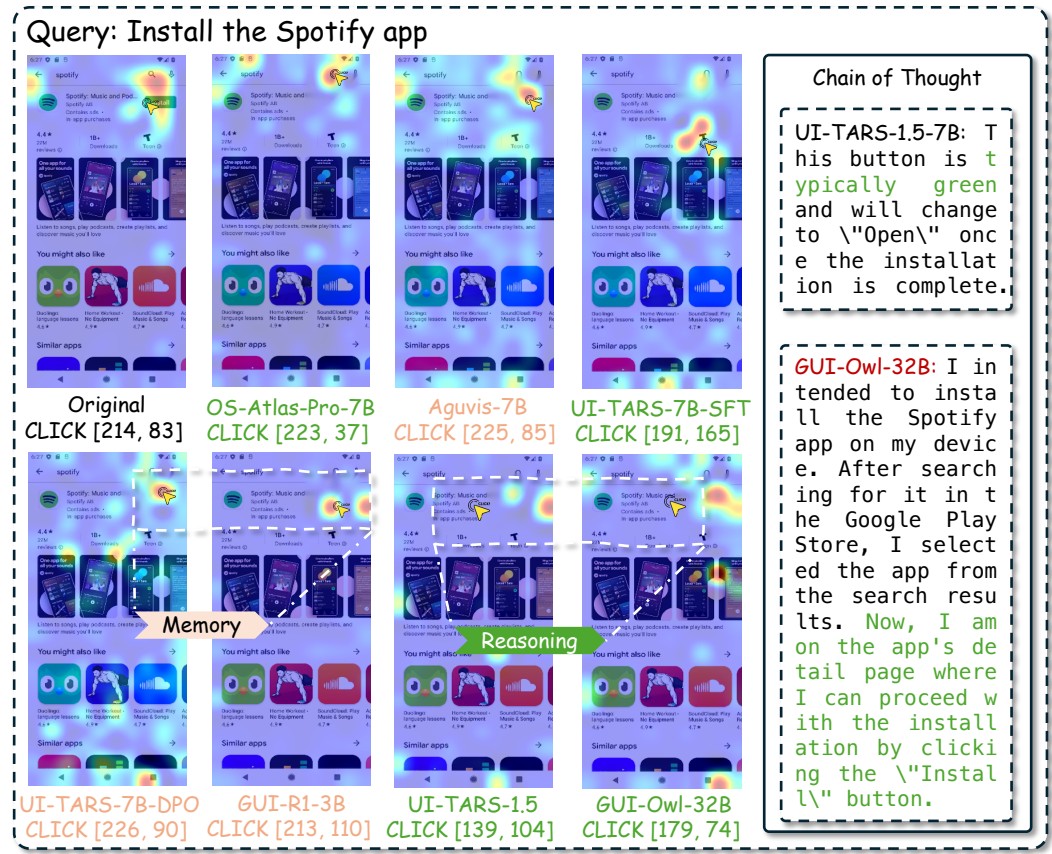

Figure 11: Visualization of relative attention in middle-layer during the decision-making process of Qwen-VL-based multimodal agents in the object editing probing. The right-side illustrate the reasoning trajectory with an integrated CoT agents.

textual content). Table 14 shows that prediction changes are predominantly driven by verb tokens, indicating that masking action verbs produces the largest degradation in agent performance.

On the one hand, masking verbs induce substantially larger SR degradation than masking objects, highlighting that most GUI agents rely heavily on the presence of explicit action verbs to reconstruct the correct atomic instruction. This effect is particularly pronounced for OS-Genesis-7B, GUI-R1-7B, and AgentCPM-GUI-8B, where removing verbs leads to increases in $\Delta P_{\text{SR}}$ to 40%~57%, whereas masking the object leads to minor changes. On the other hand, masking objects produces relative small degradation in both $\Delta P_{\text{Type}}$ and $\Delta P_{\text{SR}}$, suggesting that object tokens can be inferred precisely instead of internal memory. Therefore, the verb token emerges as the dominant driver of $\Delta P_{\text{SR}}$ sensitivity. This indicates that current GUI agents depend on instruction templates of verb and object combined than on visually grounded reasoning.

## B.4 FURTHER RESULTS OF STRUCTURE-GUIDED PROBING EXPERIMENTS

### B.4.1 COMPARISON OF CLOSED- AND OPEN- SOURCE MODELS

To comprehensively understand how closed-source models behave under structural-guided probing, we also extend our memory shortcut analysis to include three closed-source multimodal agents and compare them directly with open-source agents on the AITZ benchmark. As shown in Table 15, closed-source models exhibit consistently strong reliance on action shortcuts, particularly for PRESS and COMPLETE, where GPT-4o and Claude-4-Sonnet reach 85%~95% shortcut adherence. This suggests that even highly capable proprietary models predominantly follow atomic instruction-driven decision. However, unlike open-source agents, closed-source models also maintain high accuracy on

Table 12: Comparison of agents with and without prompt-based reflective regularization under object masking and editing, where red cells indicate memory degradation and green cells indicate no observable effect.

| GUI Agents | Methods | Object Masking | | | | Object Editing | | | |
|---|---|---|---|---|---|---|---|---|---|
| | | $\Delta P_{\text{Type}} \uparrow$ | $\Delta P_{\text{SR}} \uparrow$ | VMC↓ | RS↑ | $\Delta P_{\text{Type}} \uparrow$ | $\Delta P_{\text{SR}} \uparrow$ | VMC↓ | RS↑ |
| **Supervised Fine-Tuning** | | | | | | | | | |
| OS-ATLAS-Pro-7B | Base | 19.2 | 54.6 | 26.6 | 17.8 | 17.4 | 56.0 | 25.2 | 16.1 |
| | Prompt | 24.2 | 57.0 | 25.1 | 22.7 | 20.7 | 57.8 | 23.8 | 19.4 |
| OS-Genesis-7B | Base | 0.99 | 2.03 | 96.1 | 0.39 | 0.73 | 2.76 | 96.4 | 0.52 |
| | Prompt | 4.47 | 7.29 | 90.1 | 3.53 | 4.40 | 6.90 | 90.2 | 3.04 |
| UI-TARS-7B-SFT | Base | 1.40 | 28.4 | 64.8 | 0.99 | 1.88 | 33.7 | 58.5 | 1.19 |
| | Prompt | 1.60 | 28.5 | 64.5 | 1.15 | 2.16 | 33.8 | 58.8 | 1.35 |
| **Reinforcement Learning** | | | | | | | | | |
| GUI-R1-7B | Base | 12.0 | 45.3 | 43.1 | 11.5 | 11.6 | 45.3 | 40.8 | 10.6 |
| | Prompt | 17.4 | 47.6 | 40.7 | 16.0 | 17.1 | 47.9 | 39.2 | 15.3 |
| AgentCPM-GUI-8B | Base | 29.6 | 51.7 | 33.2 | 22.8 | 23.2 | 41.9 | 40.3 | 22.1 |
| | Prompt | 29.6 | 49.7 | 33.5 | 27.4 | 23.2 | 42.3 | 39.8 | 22.1 |
| GUI-Owl-7B | Base | 30.9 | 60.7 | 35.4 | 29.4 | 29.3 | 58.1 | 36.6 | 27.7 |
| | Prompt | 39.1 | 67.1 | 30.5 | 36.7 | 37.2 | 62.1 | 35.6 | 35.2 |

Table 13: Comparison of textual-guided probing between closed- and open-source agents at the token and sentence levels, where arrow directions reflect the ability to infer atomic instructions (token level) or ignore erroneous commands for visually grounded reasoning (sentence level).

| GUI Agents | Token-level | | Sentence-level | |
|---|---|---|---|---|
| | $\Delta P_{\text{Type}} \downarrow$ | $\Delta P_{\text{SR}} \downarrow$ | $\Delta P_{\text{Type}} \downarrow$ | $\Delta P_{\text{SR}} \downarrow$ |
| **Closed-source Models** | | | | |
| GPT-4o | 24.0 | 25.3 | 85.8 | 90.9 |
| GLM-4.5V | 11.2 | 51.0 | 24.5 | 56.7 |
| Claude-4-Sonnet | 12.7 | 14.3 | 67.9 | 84.5 |
| **Open-source Models** | | | | |
| OS-ATLAS-Pro-7B | 2.04 | 41.7 | 0.34 | 0.68 |
| OS-Genesis-7B | 37.0 | 61.0 | 53.8 | 69.4 |
| UI-TARS-7B-SFT | 3.25 | 27.7 | 38.2 | 59.7 |
| GUI-R1-7B | 44.0 | 66.0 | 96.8 | 99.4 |
| AgentCPM-GUI-8B | 4.23 | 40.7 | 33.5 | 96.9 |
| GUI-Owl-7B | 5.45 | 56.7 | 52.6 | 99.7 |

visual shortcut probing, indicating a strong ability to retain reasoning performance even when guided primarily by visual modality and high-level instructions.

Open-source models, by contrast, exhibit a far more polarized memory shortcuts. For instance, OS-Genesis-7B relies minimally on shortcuts, whereas RL/CoT-enhanced models such as AgentCPM-GUI and GUI-Owl-7B heavily exploit atomic-instruction shortcuts while showing weak robustness to visual shortcuts. This divergence indicates that shortcut dependence is driven by training data composition and alignment strategies. Hence, existing state-of-the-art GUI agents as instruction followers, behave in poor visual perception and suboptimal RS performance. This issue becomes

Table 14: Comparison of token-level textual-guided probing when masking verbs versus masking objects. Arrows directions reflect indicate the desired direction reflect the ability to infer atomic instructionsof change. Green cells highlight the masking strategy under which the model preserves action type prediction more stably, while red cells indicate the strategy under which the model maintains SR stability more reliably.

| GUI Agents | Token-level w/o verb | | Token-level w/o object | |
|---|---|---|---|---|
| | $\Delta P_{\text{Type}} \downarrow$ | $\Delta P_{\text{SR}} \downarrow$ | $\Delta P_{\text{Type}} \downarrow$ | $\Delta P_{\text{SR}} \downarrow$ |
| **Supervised Fine-Tuning** | | | | |
| OS-ATLAS-Pro-7B | 3.90 | 14.8 | 2.81 | 8.00 |
| OS-Genesis-7B | 30.5 | 57.1 | 5.29 | 21.1 |
| UI-TARS-7B-SFT | 3.40 | 34.8 | 1.14 | 12.4 |
| **Reinforcement Learning** | | | | |
| GUI-R1-7B-SFT | 13.9 | 53.2 | 5.52 | 35.2 |
| AgentCPM-GUI-8B | 1.90 | 40.8 | 0.57 | 33.0 |
| GUI-Owl-7B | 4.97 | 49.9 | 5.29 | 34.4 |

Table 15: Comparison of structure-guided probing between closed- and open-source agents on visual and action shortcuts. Highlighted values ($\geq 80\%$) indicate stronger shortcut reliance; green denotes action shortcuts and red denotes visual shortcuts. Values represent $1 - \Delta P_{\text{SR}}$.

| GUI Agents | Visual Shortcuts | | | Action Shortcuts | | |
|---|---|---|---|---|---|---|
| | SCROLL | PRESS | COMPLETE | SCROLL | PRESS | COMPLETE |
| **Closed-source Models** | | | | | | |
| GPT-4o | 54.6 | 90.5 | 73.2 | 54.1 | 89.6 | 78.5 |
| GLM-4.5V | / | 96.5 | 94.3 | / | 98.7 | 94.0 |
| Claude-4-Sonnet | 91.1 | 86.9 | 89.8 | 92.4 | 95.5 | 89.0 |
| **Open-source Models** | | | | | | |
| OS-ATLAS-Pro-7B | 97.6 | 99.3 | 99.1 | 67.0 | 86.8 | 53.6 |
| OS-Genesis-7B | 13.1 | 6.66 | 2.52 | 6.25 | 10.0 | 0.00 |
| UI-TARS-7B-SFT | 9.18 | 1.90 | 61.3 | 82.6 | 100 | 100 |
| GUI-R1-7B-SFT | 15.6 | 0.00 | 0.00 | 100 | 0.00 | 100 |
| AgentCPM-GUI-8B | 49.9 | 31.3 | 82.2 | 90.3 | 88.3 | 75.8 |
| GUI-Owl-7B | 1.06 | 3.50 | 60.8 | 0.00 | 95.9 | 98.3 |

particularly critical in autonomous execution scenarios, where over-reliance on atomic instructions can amplify the impact of malicious or erroneous commands. Our results further show that only when model capacity approaches the scale of proprietary systems do GUI agents may demonstrate reliable visual reasoning under structured action settings. These findings underscore the need for future GUI agents to tightly integrate visual perception with state-aware reasoning, rather than depending on shortcut-driven policies.

### B.4.2 CASE STUDY FOR STRUCTURE-GUIDED PROBING

Figure 12 illustrates examples of visual- and action- shortcuts in the structural-guided probing module. In visual shortcuts, we hypothesize that these reflective actions memorize specific screen information, such as loading bars during waiting periods. Consequently, the model completes tasks without instructions. In action shortcuts, we hypothesize that these reflective actions require no visual involvement and complete tasks solely through instructions. The results show that most models

Query: I want to look for a flight from Detroit to Las Vegas in business class for 4 passengers on Expedia departing October 11, 2023 and returning October 16, 2023 because I'm organising a family vacation to Las Vegas.

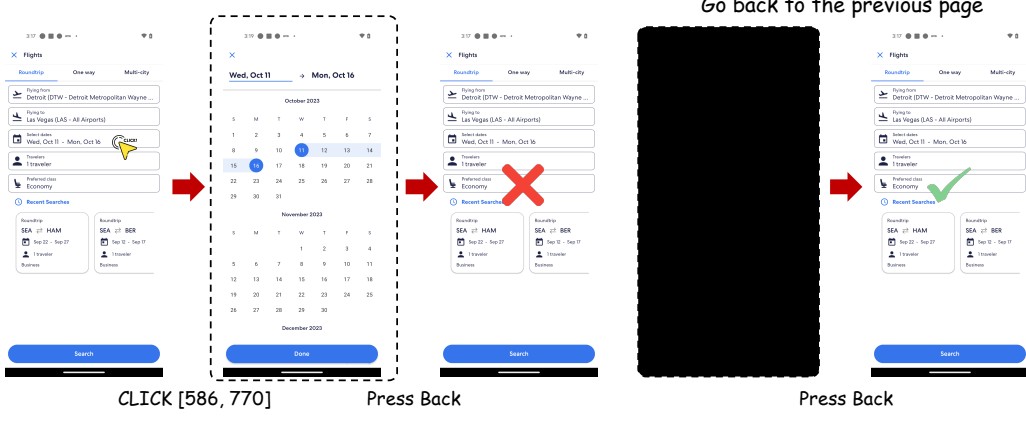

CLICK [586, 770]    Press Back       Press Back

(a) Visual Shortcuts without atomic instructions  (b) Action Shortcuts without visual modal

Figure 12: Case study of two probing methods in the structural-guided probing module.

bypass visual processing and rely on action shortcuts driven by activation of atomic commands to accomplish tasks instead of reasoning through visual perception in a genuine way.

