# OpenReview forum: "Agent-ScanKit: Unraveling Memory and Reasoning of Multimodal Agents via Sensitivity Perturbations"
_ICLR.cc/2026/Conference — Submitted to ICLR 2026_

### Official Review · Reviewer_f44Q · 2025-10-27

**Soundness:** 2
**Presentation:** 1
**Contribution:** 3
**Rating:** 4
**Confidence:** 3

**Summary:**

This paper investigates a critical problem: the unreliability of modern multimodal GUI agents, questioning whether their apparent "reasoning" is merely "spurious reasoning" or sophisticated memorization. The authors propose Agent-ScanKit, a systematic probing framework designed to dissect agent behavior by introducing controlled perturbations. By applying this toolkit to 18 agents across 5 benchmarks, the paper's primary finding is that most agents are "memory-dominated." They rely heavily on memorized spatial priors and instruction patterns rather than generalizable reasoning, which explains their brittleness on out-of-domain tasks.

**Strengths:**

- The question of "reasoning vs. memorization" is fundamental to the progress of autonomous agents. This paper provides a valuable, critical perspective on the (lack of) reliability in state-of-the-art multimodal agents.
- The specific perturbations are well-conceived. The distinction between "masking" (to test spatial memory) and "zoom-in" (to break spatial memory and test local reasoning) is particularly insightful.

**Weaknesses:**

- The paper is difficult to read. The presentation of the complex methodology, especially the distinctions between various probing outcomes (e.g., high vs. low $\Delta P_{Type}$ vs. $\Delta P_{SR}$  and the interplay with VMC and RS), is dense and difficult to track. The numerous figures and tables contain highly specific metrics and model names, requiring very careful reading to connect the data back to the core memory/reasoning concepts.
- The paper defines reasoning as non-memorized behavior under perturbation (contextual reasoning, generalization). However, distinguishing rote textual memorization (e.g., instruction adherence) from simple reasoning mechanisms remains difficult, especially in the context of the token-level versus sentence-level probing results. For instance, if an agent adheres to an erroneous instruction (sentence-level perturbation), it is labeled as instruction adherence, but the distinction between this being a "memory error" versus a failure of semantic reasoning is not fully disambiguated in the context of the GUI task flow.

**Questions:**

- Could the authors please provide a concrete, step-by-step example of how the "visual shortcut" and "action shortcut" probes are implemented?
- Can the authors provide a clearer explanation of how the quantitative metrics, particularly VMC and RS (Visual Memory Consistency and Reflection Score), directly translate into definitions of memory-dominated versus reasoning-driven behavior, especially regarding "over-reflection"? A formal definition of what constitutes "genuine reasoning" within the perturbed POMDP framework would enhance understanding.

---

> ### Author Response · Authors · 2025-11-27
>
> # Dear Reviewer f44Q
>
> > W1: The paper is difficult to read. The presentation of the complex methodology, especially the distinctions between various probing outcomes (e.g., high vs. low $\Delta P_{\text{Type}}$ vs. $\Delta P_{\text{SR}}$ and the interplay with VMC and RS), is dense and difficult to track. The numerous figures and tables contain highly specific metrics and model names, requiring very careful reading to connect the data back to the core memory/reasoning concepts.
>
> **Reply:** Thank you for the valuable feedback. We appreciate the concern regarding readability and have clarified the manuscript substantially. In the revised version, we reorganize the methodological presentation, explicitly state the purpose of each probing setup, and strengthen figure/table captions to make the connection between metrics and the underlying memory/reasoning behaviors more transparent. Concretely:
>
> 1. **Clearer explanation of the three probing frameworks:** we reorganized Section 4 to more clearly highlight the logic of Agent-ScanKit, which consists of three intuitive probing dimensions:
>    * Visual-guided probing, which separates visual memory (masking/editing) from visual reasoning (Zoom-in).
>    * Text-guided probing, which analyzes text-content predictions within Type actions, including (i) fine-grained verb–argument mapping, and (ii) reasoning under erroneous instructions.
>    * Structure-guided probing, which examines whether reflective actions—rare in training—have formed visual or action shortcuts.
>
> Each probing type is now clearly labeled with its purpose and expected behavioral signatures.
>
> 2. **Clearer interpretation of the four core metrics:** We clarified that the unified metric $\Delta P$ captures sensitivity to perturbation across all probing types:
>
>    * In visual-guided probing, Low $\Delta P$: memory-dominant behavior in masking/editing, and reasoning success in Zoom-in; VMC measures visual prediction consistency; higher values indicate stronger positional memory; RS measures reflective behavior; higher values indicate detected anomalies and corrective actions.
>    * In text-guided probing, $\Delta P$ reflects whether agents rely on atomic instruction templates or can perform visual-textual joint reasoning under incorrect instructions.
>    * In structure-guided probing, we additionally use 1-\Delta P to directly capture “shortcut-induced correctness” when vision is removed or atomic instruction is removed.
> 3. **Improved presentation linking data to core concepts:** We revised figure captions, added explanatory arrows for metric directions, and expanded the narrative to explicitly connect each result to the paper’s central concepts of memory vs. reasoning. The results are now discussed under a unified framework showing how differences in the reasoning behaviour of various models arise under different training paradigms, model scales, modal ablation, and threshold settings.
>
> Overall, we have made every effort to improve the presentation quality within a short timeframe so that readers can better understand the contributions of this work.

---

> ### Author Response · Authors · 2025-11-27
>
> > W2: The paper defines reasoning as non-memorized behavior under perturbation (contextual reasoning, generalization). However, distinguishing rote textual memorization (e.g., instruction adherence) from simple reasoning mechanisms remains difficult, especially in the context of the token-level versus sentence-level probing results. For instance, if an agent adheres to an erroneous instruction (sentence-level perturbation), it is labeled as instruction adherence, but the distinction between this being a "memory error" versus a failure of semantic reasoning is not fully disambiguated in the context of the GUI task flow.
>
> **Reply:** Thank you for raising this important conceptual concern. We agree that disentangling rote textual memorization from genuine semantic reasoning is challenging, especially in multimodal GUI settings. Our analysis, however, combines text-guided probing and structure-guided probing to clearly separate these two behaviors.
>
> 1. First, the token-level text probing demonstrates that models rely heavily on atomic-instruction memorization rather than semantic interpretation. As shown in Table 2, most agents fail to infer the correct textual content when the verb is masked. This reveals a strong dependence on a fixed mapping between atomic verbs (e.g., input, type) and predicted action content. Our ablation further shows that masking the object (the argument of the verb) barely affects the model’s predictions, confirming that the core memory lies in a verb-driven action template, not in understanding the sentence semantics.
> 2. Second, the sentence-level perturbation exposes a deeper separation between memory and reasoning. When the entire instruction is replaced with an incorrect or contradictory one, agents still follow it regardless of the visual state. Because GUI tasks fundamentally require integrating visual cues with textual goals, blindly executing an erroneous instruction—without referencing the screen—is not a failure of reasoning, but rather a memory-triggered behavior driven by the learned atomic-instruction template. This reinforces that the agent is not performing semantic reasoning but retrieving a stored action pattern.
> 3. Third, structural-guided probing reveals the same shortcut behavior in reflective actions. While reflection should involve interpreting the screen state and correcting the task trajectory, we observe that many reflective actions are also activated directly by atomic-instruction templates, without requiring visual inspection. This is contradictory to the expected purpose of reflection, and points again to instruction-triggered memory shortcuts, not contextual reasoning.
>
> Overall, the results allows us to disambiguate these behaviors that are not ambiguous mixtures of memory and reasoning, but rather clear evidence that current multimodal GUI agents rely on atomic-instruction-triggered memory activation rather than true multimodal semantic reasoning.
>
> > Q1: Could the authors please provide a concrete, step-by-step example of how the "visual shortcut" and "action shortcut" probes are implemented?
>
> **Reply:** Thanks for the suggestion. We have added detailed descriptions in Section 4.3 and Appendix B.4.2. Below is a concise, step-by-step summary of how the visual shortcut and action shortcut probes are implemented.
>
> 1. We select the steps where all agents successfully executed reflective actions under standard settings.
> 2. Visual shortcut probing: we keep the original screenshot and high-level task instruction but remove all atomic instructions, then ask the agent for its next action. If it still succeeds, the agent may rely on visual shortcuts.
> 3. Action shortcut probing: we replace the screenshot with a fully black image, keep the original atomic instructions, and query the agent again. If it still succeeds, the agent may rely on instruction-only shortcuts.
>
> These controlled ablations isolate whether reflective success depends on visual patterns, atomic instructions, or genuine multimodal integration.

---

> ### Author Response · Authors · 2025-11-27
>
> > Q2: Can the authors provide a clearer explanation of how the quantitative metrics, particularly VMC and RS (Visual Memory Consistency and Reflection Score), directly translate into definitions of memory-dominated versus reasoning-driven behavior, especially regarding "over-reflection"? A formal definition of what constitutes "genuine reasoning" within the perturbed POMDP framework would enhance understanding.
>
> **Reply:** Thank you for the insightful question. We add clarifications in Section 4.1 and Appendix A.5. Below we provide a concise explanation of how VMC and RS connect to memory dominated and reasoning driven behavior, and how genuine reasoning is defined in our perturbed POMDP setting.
>
> 1. Perturbed POMDP definition of reasoning and memory. We evaluate only trajectories that the agent originally solves correctly. Under the perturbed POMDP, we redefine Eq. (3) to measure the performance change, and we use this metric consistently across all probing methods.
>
>    * Small $\Delta P$ in visual guided probing represent strong visual memory on masking and editing probing, and visual reasoning on zoom in.
>    * Small $\Delta P$ in textual guided probing represent strong instruction inference reasoning at the token level, and visual reasoning at the sentence level.
>    * In the structure guided probing, we report $1-\Delta P$ for clearer interpretation that higher values represent stronger shortcut reliance.
>
> 2. How VMC and RS map to memory dominated and reasoning driven behavior. First, VMC, which is Visual Memory Consistency, is the prediction consistency before and after visual perturbation. High VMC represent that the action distribution of the model is unchanged and that the model relies strongly on memorized visual patterns, which is memory dominated behavior. Second, RS, which is the Reflection Score, measures how often the agent triggers reflective actions such as back tracking and terminating after perturbation. Appropriate reflection can detect abnormal states, and high RS often appear when global visual corruption occurs (Zoom-in) rather than real anomaly detection. This indicates instability rather than genuine reasoning.
>
> Overall, under the perturbed POMDP formulation, memory dominated behavior corresponds to high VMC, and reasoning driven behavior is characterized by small $\Delta P$, low VMC, and stable RS without over reflection.

---

### Official Review · Reviewer_xwUw · 2025-10-31

**Soundness:** 3
**Presentation:** 3
**Contribution:** 3
**Rating:** 8
**Confidence:** 3

**Summary:**

This paper introduces Agent-ScanKit, a systematic probing framework designed to analyze and quantify the contributions of memorization versus genuine reasoning in multimodal agents interacting with Graphical User Interfaces (GUIs). Motivated by the observation that existing agents exhibit poor reliability on complex or out-of-domain tasks, the framework uses sensitivity perturbations across three orthogonal paradigms—visual-guided, text-guided, and structure-guided—to isolate and measure reliance on memory shortcuts. Evaluating 18 agents on five GUI benchmarks, the authors demonstrate that mechanical memorization often outweighs systematic reasoning. The findings suggest that many current models function primarily as sophisticated retrievers of training-aligned knowledge, leading to limited generalization and unreliable behavior, underscoring the urgent need for better reasoning mechanisms.

**Strengths:**

1. The introduction of three orthogonal probing paradigms (Visual-guided, Text-guided, and Structure-guided) offers a comprehensive way to dissect agent behavior along the entire input-to-output pipeline (perception, instruction, action selection).
2. The paper provides compelling quantitative proof of memory shortcuts. For instance, the high VMC (Visual Memory Consistency) and persistently low $\mathbf{\Delta P_{Type}}$ under visual perturbations strongly suggest reliance on memorized spatial priors (Table 1).
3. The structure-guided probing highlights concrete issues with state and reflection actions (e.g., poor performance on PRESSBACK/WAIT and reliance on Action Shortcuts in WAIT/COMPLETE), giving developers clear targets for improving the action space and reflection mechanisms.
4. Large-Scale Validation: Evaluating a large and diverse set of 18 state-of-the-art open-source agents across multiple benchmarks enhances the generalizability of the conclusions.

**Weaknesses:**

1. The paper uses high sensitivity ($\mathbf{\Delta P}$ is large) as evidence for memory dominance and low sensitivity ($\mathbf{\Delta P}$ is small) as evidence for reasoning/robustness. However, a small $\mathbf{\Delta P}$ under simple perturbations could also signify memorization (the model robustly retrieves the memorized item despite minor noise), while a large $\mathbf{\Delta P}$ under complex OOD perturbations (like zoom-in) could be attributed to a brittle policy that fails to generalize, rather than just "memory." The discussion could be strengthened by acknowledging this nuance and providing a clearer definition of where "memory" ends and "brittle policy" begins.

2. The visual-guided probing (Table 1) mixes visual input (screenshot) with optional text input (low-level instruction). A crucial ablation is missing: how does the dependency on visual memory (low $\mathbf{\Delta P_{Type}}$) change when the textual input (AX Tree or instruction) is removed or perturbed first? This would isolate the true contribution of visual memory versus memory formed by the visual-textual rule matching.

3. The paper notes RL agents exhibit stronger reflective capability but sometimes introduces side effects. A more focused discussion is needed on why RL—a paradigm often associated with generalization—still struggles with memory bias. Is the RL reward signal ($R$) insufficient to penalize memory retrieval, or is the environment ($T$) too visually/textually uniform to demand true OOD reasoning?

**Questions:**

1. Please clarify the role of the supplementary data (e.g., text representations like AX Tree) in the visual-guided probing (Table 1). Specifically, if an agent uses both the screenshot and the low-level instruction, what happens if the screenshot is masked but the instruction is left unperturbed? How does the model perform its action type prediction, and how does this performance compare to the visual-only results presented?

2. The paper suggests that RL agents' reflexivity introduces notable side effects. For models that rely on CoT (e.g., AgentCPM-GUI), can the authors provide a brief quantitative analysis of the cost (e.g., average token count or latency increase) associated with the reflexivity observed under perturbation, compared to normal execution?

3. In the zoom-in probing (Table 1), OS-Genesis-7B shows $\mathbf{\Delta P_{SR}}$ of $\mathbf{98.8\%}$ and $\mathbf{VMC}$ of $\mathbf{86.0\%}$ (high memory consistency), yet $\mathbf{\Delta P_{Type}}$ is low at $\mathbf{2.9\%}$. This suggests the model correctly predicts the action type (e.g., CLICK) but consistently fails the grounding/SR check. Is this because the spatial memory for the target location is disrupted by the crop/zoom, or because the zoom removes vital global context necessary for accurate localization?

---

> ### Author Response · Authors · 2025-11-27
>
> # Dear Reviewer xwUw
>
> **Thanks for your insightful review and constructive feedback.**
>
> > W1: The paper uses high sensitivity (is large) as evidence for memory dominance and low sensitivity (is small) as evidence for reasoning/robustness. However, a small under simple perturbations could also signify memorization (the model robustly retrieves the memorized item despite minor noise), while a large under complex OOD perturbations (like zoom-in) could be attributed to a brittle policy that fails to generalize, rather than just "memory." The discussion could be strengthened by acknowledging this nuance and providing a clearer definition of where "memory" ends and "brittle policy" begins.
>
> **Reply:** Thank you for raising this important conceptual nuance. We clarify that the memory probes apply localized, controlled perturbations using a fixed 50$\times$50 pixel mask or edit that removes only the target while preserving the global layout. In this setting, small values of $\Delta P_{Type}$ or $\Delta P_{SR}$ indicate memory because the correct visual cues are removed, yet the model still produces the same prediction. To define more precisely where memory ends, we include the following points:
>
> 1. Masking and editing ratio ablations that range from 10 percent to 100 percent show how sensitivity increases once memory no longer compensates for missing visual evidence. As the ratio increases, we observe that models consistently default to memory driven prediction even under extreme masking or editing. Scaling perturbation ratios has minimal effect on memory biased models. RL or CoT models show greater SR sensitivity and re-exploration than SFT models. However, certain models such as OS Genesis remain almost entirely position memory regardless of perturbation strength. More details appear in Table 10 of Appendix B.2.5.
>
> 2. Ablation of visual modality and atomic instruction defines the upper bound of absolute memory, in which no visual or atomic instruction is available. When the visual modality and the atomic instruction are introduced, they undertake different roles: SFT models show increased memory and RL models show memory degradation. The details appear in Figure 7 of Appendix B.2.3.
>
> Meanwhile, we agree that zoom in is intentionally out of distribution and measures reasoning and generalization rather than memory. Our intention is to quantify reasoning stability when the global context is reduced but the target remains visible. To strengthen this interpretation, we report quadrant 25 percent zoom in versus horizontal 50 percent zoom in, which highlights that models degrade more under stronger out of distribution conditions and reveals model specific differences in reasoning stability. More details appear in Table 11 of Appendix B.2.5.
>
> These analyses define cases in which low sensitivity reflects memorized policy execution under memory probing, and high sensitivity under global context reducing out of distribution perturbations indicates brittle reasoning and generalization.

---

> ### Author Response · Authors · 2025-11-27
>
> > W2&Q1: The visual-guided probing (Table 1) mixes visual input (screenshot) with optional text input (low-level instruction). A crucial ablation is missing: how does the dependency on visual memory (low $\Delta P_{Type}$) change when the textual input (AX Tree or instruction) is removed or perturbed first? This would isolate the true contribution of visual memory versus memory formed by the visual-textual rule matching.
>
> **Reply:** Thank you for the insightful suggestion. We agree that disentangling the contributions of visual and textual modalities is essential for determining whether low $\Delta P_{\text{Type}}$ reflects genuine visual memory or merely visual–textual rule matching. To address this concern, we have added the requested ablation study. Specifically, we evaluate four settings: (i) without both visual input and atomic instructions, (ii) with atomic instructions but no visual input, (iii) with visual input but no atomic instructions, and (iv) with both visual input and atomic instructions. Table W2 reports the resulting $\Delta P_{\text{Type}}$ values. Our key findings are as follows:
>
> **Table W2: Ablation study of different settings in the object-masking of visual-guided probing. ``Vis‘’ = Visual Modality, ``Instr'' = Atomic Instruction. ($\Delta P_{Type}$)**
>
> | Model           | w/o Vis+Instr | w/o Vis | w/o Instr | w/ Instr |
> | --------------- | ------------- | ------- | --------- | -------- |
> | OS-ATLAS-Pro-7B | 29.9          | 25.7    | 14.0      | 13.0     |
> | OS-Genesis-7B   | 23.1          | 15.3    | 7.42      | 1.11     |
> | UI-TARS-SFT     | 94.6          | 21.0    | 18.7      | 10.7     |
> | GUI-R1-7B       | 47.6          | 35.3    | 12.7      | 13.1     |
> | AgentCPM-GUI-8B | 50.2          | 50.2    | 9.32      | 29.2     |
> | GUI-Owl-7B      | 43.7          | 40.0    | 9.17      | 15.3     |
>
> 1. No-visual settings. Atomic instructions substantially reduce $\Delta P_{\text{Type}}$ for UI-TARS-7B-SFT and slightly reduce it for other models, reflecting visual–textual rule matching. However, SFT models inherently exhibit lower $\Delta P_{\text{Type}}$. In contrast, RL models show consistently higher $\Delta P_{\text{Type}}$. This indicates that visual input is essential for activating the agent’s memory.
>
> 2. Visual-available settings but without atomic instructions (before masking/editing). Removing atomic instructions has almost no effect on $\Delta P_{\text{Type}}$, which remains low by default (e.g., 9.2\%$\sim$12.7% for RL models and 7.4\%$\sim$18.7% for SFT models). Once atomic instructions are introduced, SFT models further reinforce memory-driven predictions, whereas RL models tend to activate reflective behavior, resulting in increased $\Delta P_{\text{Type}}$.
>
> Overall, the supplemeted ablations show that (i) visual input is the dominant source of positional memory; (ii) atomic instructions play fundamentally different roles in SFT vs. RL models; and (iii) current multimodal agents exhibit unreliable reasoning even when textual input is removed. The full results are provided in Appendix B.2.3 (Figures 7 and 8).

---

> ### Author Response · Authors · 2025-11-27
>
> > W3: The paper notes RL agents exhibit stronger reflective capability but sometimes introduces side effects. A more focused discussion is needed on why RL—a paradigm often associated with generalization—still struggles with memory bias. Is the RL reward signal (R) insufficient to penalize memory retrieval, or is the environment (T) too visually/textually uniform to demand true OOD reasoning?
>
> **Reply:** Thank you for the thoughtful question. In general, RL agents indeed exhibit stronger reflective capability, particularly in visual-memory probing, where they consistently show higher $\Delta P_{\text{Type}}$ and RS. However, they still suffer from pronounced memory bias. Our additional analyses suggest three underlying reasons.
>
> 1. RL-based agents also rely on global visual structure learned during training. In the Zoom-in setting, where global visual structure is disrupted, RL agents thus produce high $\Delta P_{\text{Type}}$ and RS. This indicates that reflection is not triggered by genuine reasoning, but by a mismatch between the expected global layout and the perturbed input. In other words, RL models appear to have learned a layout-dependent policy. Once the global structure becomes unfamiliar, reflection is activated as a side effect rather than as principled reasoning.
>
> 2. RL reward signals tend to reinforce instruction-following rather than visual-grounded reasoning. In textual-guieded probing, we observe that RL agents often fail to interpret atomic instructions meaningfully and instead execute them blindly. This suggests that both SFT initialization and RL fine-tuning encourage agents to rely on the format of atomic instructions rather than on visual evidence. Thus, the RL objective inadvertently strengthens instruction adherence while failing to penalize memory-driven shortcuts or hallucinated visual inference.
>
> 3. The training environments are visually and textually uniform, which weakens the incentive to develop OOD reasoning. Because most GUI environments share consistent structural layouts, RL agents can optimize reward by forming shallow action shortcuts without relying on perceptual reasoning. This further amplifies memory bias. gents can retrieve memorized spatial priors to solve tasks at training time, without ever needing to infer from visual changes.
>
> Overall, current RL-based GUI agents behave more like structured instruction executors than multimodal planners or reasoners. They do not yet acquire a true understanding of GUI manipulation logic, and the dominance of memory-driven shortcuts limits their ability to generalize beyond familiar visual layouts. This explains why RL, despite its reputation for improving generalization, still struggles with memory bias in multimodal GUI agents.
>
> > Q2: The paper suggests that RL agents' reflexivity introduces notable side effects. For models that rely on CoT (e.g., AgentCPM-GUI), can the authors provide a brief quantitative analysis of the cost associated with the reflexivity observed under perturbation, compared to normal execution?
>
> **Reply:** Thank you for the question. We first clarify that high RS values reflect over-reflection, where RL/CoT-based agents unnecessarily execute reflective actions under Zoom-in perturbations because the global visual structure is disrupted, even though they should directly perform visual localization.
>
> **Table Q2: Average number of generated CoT tokens for each model under different perturbation settings.**
>
> | Model          | Original | Object Masking | Object Editing | Zoom-in |
> | -------------- | -------- | -------------- | -------------- | ------- |
> | GUI-Owl-7B     | 50.47    | 52.69          | 52.26          | 47.10   |
> | GUI-Owl-32B    | 51.51    | 52.7454        | 52.20          | 47.41   |
> | AgentCPM-GUI   | 15.39    | 15.4790        | 15.47          | 15.57   |
> | UI-TARS-7B-SFT | 11.12    | 10.7126        | 10.69          | 10.79   |
>
> To quantify the computational cost associated with this behavior, we report in Table Q2 the average number of generated CoT reasoning tokens under perturbation. As shown, the token counts remain nearly unchanged across all conditions. This indicates that the reflexivity observed in these agents does not lead to a meaningful increase in inference verbosity or reasoning-chain length.
>
> Moreover, our visualization results (Figures 10–11 in Appendix B.2.6) shows that the generated CoT traces are almost identical across different perturbations. This suggests that the CoT sequences produced by current RL/CoT-based GUI agents behave as superficial, template-like reasoning, failing to account for visual anomalies or the underlying perceptual changes.
>
> Overall, although RL agents exhibit stronger reflective behavior, their CoT does not adapt to visual perturbations and contributes little beyond fixed reasoning patterns. This highlights that their reflexivity is structural rather than perceptual, and does not incur additional token-generation costs.

---

> ### Author Response · Authors · 2025-11-27
>
> > Q3: In the zoom-in probing (Table 1), OS-Genesis-7B shows $\Delta P_{SR}$ of 98.8 and VMC of 86.6 (high memory consistency), yet $\Delta P_{Type}$ is low at 2.9. This suggests the model correctly predicts the action type (e.g., CLICK) but consistently fails the grounding/SR check. Is this because the spatial memory for the target location is disrupted by the crop/zoom, or because the zoom removes vital global context necessary for accurate localization?
>
> **Reply:** Thank you for the instructive question. We would like to clarify the behavior of OS-Genesis-7B under the Zoom-in probing. In the Zoom-in setting, the target region is still present, but the global visual structure is deliberately disrupted to evaluate the agent’s ability to perform local visual reasoning. The extremely high VMC indicates that the model retains strong positional memory, where its predictions remain highly consistent before and after the perturbation. Thus, the model is unable to inference the target’s location after zooming, leading to the very high $\Delta P_{\text{SR}}$ observed. In other words, OS-Genesis-7B continues to click its memorized coordinates rather than re-ground the target under the altered visual scale.

---

### Official Review · Reviewer_ct7c · 2025-11-01

**Soundness:** 2
**Presentation:** 2
**Contribution:** 2
**Rating:** 4
**Confidence:** 2

**Summary:**

The paper introduces Agent-ScanKit, a black-box probing framework for GUI agents that diagnoses whether success arises from memorization or genuine reasoning. It defines three orthogonal perturbation families—visual-guided, text-guided, and structure-guided—and evaluates 18 open-source multimodal agents across five public GUI benchmarks. Findings: many agents rely heavily on mechanical memorization.

**Strengths:**

1. Originality. Clear, orthogonalized probing design (visual / text / structure) that goes beyond final-answer or step-wise accuracy to characterize where agents lean on memory vs. reasoning. The structure-guided notion of visual vs. action shortcuts is a useful conceptual lens for reflection/state actions in GUI control.

2. Quality. Concrete, interpretable perturbation deltas, plus VMC and RS for visual probes; detailed per-agent tables illustrate systematic patterns.

3. Significance. Addresses a central question for GUI agents: are gains from true reasoning or overfit priors? The toolkit offers a portable, black-box diagnostic that practitioners can run today; results suggest research should re-balance away from spatial/instruction memorization toward robust reasoning.

**Weaknesses:**

1. Causal interpretation of “reasoning” vs. “memorization”.
The paper interprets small/large delta P under certain perturbations as evidence of memory/reasoning, which is the main assumption. But delta p can also be affected by nuisance factors (e.g., crop changing saliency distribution, off-screen context removal). Lack of ablation controls (random crops vs. target-preserving crops; mask non-targets; swap layout regions) and report causal contrast analyses to isolate the specific factor each probe intends to test.

2. Metric grounding and thresholds.
VMC depends on a distance threshold (50 px); RS is introduced but its calibration isn’t deeply justified. Lack sensitivity sweeps for thresholds and show ranking stability under reasonable ranges.

3. Scope of structural probes.
“Visual/action shortcuts” are defined for a few reflection/state actions (WAIT, PRESS, COMPLETE, SCROLL); real deployments face richer failure modes (timeouts, latency spikes, theme/scale changes, language/locale, accessibility overlays). Would be good to extend structure-guided probes to system-level perturbations and tool/environment faults.

**Questions:**

1. Probe validity & confounds: For zoom-in, how do you ensure that the cropped view still preserves all necessary cues for localization/decision (e.g., loss of breadcrumb anchors)? Any human verification that tasks remain solvable under zoom-in?

2. Instruction perturbations: In token-level tests, agents degrade mainly in vocabulary prediction while Type stays stable. Can you quantify which tokens (verbs vs. objects vs. app names) drive delta PSR most?

3. Comparisons with alternative process evaluators: Could Agent-ScanKit be combined with trajectory metrics (e.g., grounded-step checks, evidence-bank style) to see whether memory-heavy agents also show more ungrounded reflections?

4. Practical guidance: Given your findings, what training interventions reduce shortcutting (e.g., layout-swap augmentation, coordinate jitter, instruction paraphrase curricula, reflection regularizers)—any preliminary results?

---

> ### Author Response · Authors · 2025-11-27
>
> # Dear Reviewer ct7c
>
> **Thanks for your insightful review and constructive feedback.**
>
> > W1: Causal interpretation of “reasoning” vs. “memorization”. The paper interprets small/large delta P under certain perturbations as evidence of memory/reasoning, which is the main assumption. But delta p can also be affected by nuisance factors (e.g., crop changing saliency distribution, off-screen context removal). Lack of ablation controls (random crops vs. target-preserving crops; mask non-targets; swap layout regions) and report causal contrast analyses to isolate the specific factor each probe intends to test.
>
> **Reply:** In the revised version, we supplement the distribution of visual field saliency under different masking or editing ratios and different zoom-in ratios (Table 10-11 and Figure 9 of Appendix B.2.5). We also conduct ablation study of visual modality and automic instruction (Figure 7 and 8 of Appendix B.2.3). Furthermore, we include masking and editing of random non-object fields alongside the retention of target visual fields (Table 9 of Appnedix B.2.4). Our insights are as follows:
>
> 1. **Masking or editing ratios and different zoom-in ratios:** stronger perturbations, focused views, or altered VMC thresholds merely rescale memory intensity. Although $\Delta P$ may exhibit fluctuations, GUI agents under uniform settings will display distinct reasoning mechanisms. For instance, RL tends to be more reflective than SFT with small position memory.
> 2. **Ablation study of visual modality and automic instruction:** visual modality is essential for grounding, with atomic instructions amplifying memory in SFT models and partially mitigating it in RL models. Moreover, while atomic instruction absence, GUI agents still exhabit memory behavior.
> 3. **Random non-object masking and editing:** perturbing non-object regions barely affects prediction. This reflects the agents’ dominant reliance on memorized coordinates.
>
> To clarify the purpose of different probes, we have highlighted their probing ranges and functions in the revised version (Lines 70-77, Page 2). The details are as follows:
>
> 1. **Visiual-guided detection:** the probing target is the visual localization task. By masking and editing target locations, this framework quantifies consistency between an agent's forward and backward predictions, and whether it engages in reflection when targets disappear. Through zoom-in settings, it assesses whether reasoning remains feasible while preserving local information, mirroring human focalized positioning.
> 2. **Text-guided probing:** the probe targets text input tasks. By perturbing atomic instructions at token and sentence levels, it verifies their intrinsic impact on textual content predictions.
> 3. **Structure-guided Probing:** as analysed in Section 3, extensive data and algorithms favour visual localisation and textual tasks, leading to state and reflective judgement actions relying on memory shortcuts. Our paper aims to quantify whether models analyse agent reasoning mechanisms through remembering visual shortcuts or atomic instructions-driven shortcuts.
>
> Overall, the three probing methods are mutually independent and do not interfere with one another, enabling comprehensive analysis of an agent's reasoning behaviour across its entire action space.
>
> > W2: Metric grounding and thresholds. VMC depends on a distance threshold (50 px); RS is introduced but its calibration isn’t deeply justified. Lack sensitivity sweeps for thresholds and show ranking stability under reasonable ranges.
>
> **Reply:** Thank you for your suggestion. In the revised version, we add a VMC threshold sensitivity analysis in **Figure 9 in Appendix B.2.5**. The results show that the VMC threshold only rescales the magnitude and saturates beyond about twenty percent, which shows that the metric is stable. Thus, we choose fifty percent, which is generally distributed around the target. The main findings are as follows:
>
> 1. **Object masking and editing:** Models retain absolute positional memory at zero percent, and VMC increases rapidly within zero to twenty percent. SFT models show strong positional memory, while RL or CoT models grow more slowly with the threshold.
> 2. **Zoom in:** With global structure disrupted, most models shift to inference regardless of the threshold, even though grounding is lower correct.
>
> In addition, we add a clear definition of the reflection score in Section 4 to clarify that it measures the proportion of reflective actions such as PressBack, PressHome, Wait, Complete and Scroll that trigger when the agent encounters anomalous behavior.

---

> ### Author Response · Authors · 2025-11-27
>
> > W3: Scope of structural probes. “Visual/action shortcuts” are defined for a few reflection/state actions (WAIT, PRESS, COMPLETE, SCROLL); real deployments face richer failure modes (timeouts, latency spikes, theme/scale changes, language/locale, accessibility overlays). Would be good to extend structure-guided probes to system-level perturbations and tool/environment faults.
>
> **Reply:** Thank you for your suggestion. We clarify that structure guided probes refer to shortcut learning in multimodal agents that arises from data distribution or algorithmic bias, including visual shortcuts and textual shortcuts. We focus on actions with low distribution that may contain potential shortcut learning, such as reflection and state actions, including WAIT, PRESS, COMPLETE and SCROLL. As shown in Section 5.2.3 three and Appendix B.4.2, most models rely on action shortcuts without participation of the visual modality, which is usually an unreliable reasoning mechanism.
>
> Meanwhile, the results from our visual-guided probing extension remain consistent with the findings in Table 1. First, **theme evaluation**, the conclusions from memory/reasoning probing on the AITZ benchmark (Table 8) align with outcomes across multiple benchmarks reported in Table 1. Second, **scale changes**, by analyzing the Zoom-in ratio (Table 11)**, we compared the reasoning behaviors of several agents under different levels of local information reduction, and observed similarly consistent trends.

---

> ### Author Response · Authors · 2025-11-27
>
> > Q1: Probe validity & confounds: For zoom-in, how do you ensure that the cropped view still preserves all necessary cues for localization/decision (e.g., loss of breadcrumb anchors)? Any human verification that tasks remain solvable under zoom-in?
>
> **Reply:** Thank you for the insightful question. We clarify below how our zoom-in probes preserve task solvability and observe how alterations in global visual structure affect the decline in reasoning ability.
>
> 1. **Target visibility is always preserved:** zoom-in can reserve the target-critical region. In the quadrant-based zoom-in setting, the crop is not arbitrary. We first locate the ground-truth bounding box of the target and then perform a four-quadrant split; the system selects only the quadrant guaranteed to contain the target (see Fig. 4). Thus, the agent always retains full visibility of the target object or UI element. The probe evaluates whether the model can still reason about the correct spatial location when visual context reduced under uniform visual setting.
>
> **Table Q1: Comparison of zoom-in probing strategies using 4-way quadrant crops and 2-way vertical splits across six multimodal GUI agents.**
>
> | **GUI Agents**             | **ΔP_Type↓ (4-way)** | **ΔP_SR↓ (4-way)** | **VMC↓ (4-way)** | **RS↓ (4-way)** | **ΔP_Type↓ (2-way)** | **ΔP_SR↓ (2-way)** | **VMC↓ (2-way)** | **RS↓ (2-way)** |
> | -------------------------------- | ---------------------------- | -------------------------- | ----------------------- | ---------------------- | ---------------------------- | -------------------------- | ----------------------- | ---------------------- |
> | **Supervised Fine-tuning** |                              |                            |                         |                        |                              |                            |                         |                        |
> | OS-ATLAS-Pro-7B                  | 13.6                         | 40.5                       | 0.72                    | 12.3                   | 5.94                         | 18.6                       | 0.28                    | 4.88                   |
> | OS-Genesis-7B                    | 2.90                         | 98.8                       | 86.0                    | 1.61                   | 5.79                         | 68.0                       | 69.4                    | 1.77                   |
> | UI-TARS-7B-SFT                   | 12.1                         | 36.5                       | 1.43                    | 8.66                   | 4.10                         | 11.8                       | 1.49                    | 1.59                   |
> | **Reinforcement Learning** |                              |                            |                         |                        |                              |                            |                         |                        |
> | GUI-R1-7B                        | 4.90                         | 44.6                       | 0.40                    | 4.00                   | 2.71                         | 34.5                       | 0.74                    | 2.11                   |
> | AgentCPM-GUI-8B                  | 14.2                         | 42.9                       | 1.31                    | 13.7                   | 4.39                         | 13.9                       | 3.27                    | 3.73                   |
> | GUI-Owl-7B                       | 8.60                         | 38.9                       | 0.36                    | 7.23                   | 5.14                         | 12.9                       | 0.42                    | 4.30                   |
>
> 2. To further validate solvability, we additionally design a horizontal-splitting zoom-in that keeps a substantially larger portion of the global layout. Table Q1 compares the two settings. We observe that horizontal segmentation leads to significantly smaller changes in $\Delta P_{Type}$, $\Delta P_{SR}$, and RS. This demonstrates that the tasks remain solvable under zoom-in, and that preserving more structural layout indeed stabilizes reasoning. Notably, OS-Genesis-7B and GUI-R1-7B show the strongest degradation under quadrant zoom-in, reflecting their reliance on global layout memory rather than robust visual reasoning.
>
> Overall, both scaling settings preserve the solubility of the task. Visual context cropping at different scales enables a more nuanced assessment of the reasoning capabilities of agents within local visual modalities. Notably, we incorporate the results and analysis in our revised paper in Appendix B.2.5.

---

> ### Author Response · Authors · 2025-11-27
>
> > Q2: Instruction perturbations: In token-level tests, agents degrade mainly in vocabulary prediction while Type stays stable. Can you quantify which tokens (verbs vs. objects vs. app names) drive delta PSR most?
>
> **Reply:** Thank you for the insightful question. To further disentangle how textual cues influence spatial reasoning, we extend our token-level analysis by separately masking verb and object words in atomic instructions. Verb masking uses the original setup, while object masking randomly masks only the object tokens (e.g., type “how to [] our paper”). Table Q2 presents the results. Our findings are as follows:
>
> **Table Q2: Comparison of token-level textual-guided probing when masking verbs versus masking objects.Arrows indicate desired direction reflecting ability to infer atomic instruction changes.**
>
> | **GUI Agents**             | **ΔP_Type↓ (w/o verb)** | **ΔP_SR↓ (w/o verb)** | **ΔP_Type↓ (w/o object)** | **ΔP_SR↓ (w/o object)** |
> | -------------------------------- | ------------------------------- | ----------------------------- | --------------------------------- | ------------------------------- |
> | **Supervised Fine-Tuning** |                                 |                               |                                   |                                 |
> | OS-ATLAS-Pro-7B                  | 3.90                            | **14.8**                | **2.81**                    | 8.00                            |
> | OS-Genesis-7B                    | 30.5                            | **57.1**                | **5.29**                    | 21.1                            |
> | UI-TARS-7B-SFT                   | 3.40                            | **34.8**                | **1.14**                    | 12.4                            |
> | **Reinforcement Learning** |                                 |                               |                                   |                                 |
> | GUI-R1-7B-SFT                    | 13.9                            | **53.2**                | **5.52**                    | 35.2                            |
> | AgentCPM-GUI-8B                  | 1.90                            | **40.8**                | **0.57**                    | 33.0                            |
> | GUI-Owl-7B                       | **4.97**                  | **49.9**                | 5.29                              | 34.4                            |
>
> 1. Degradations in $\Delta P_{SR}$ are driven almost entirely by **verbs, not objects**. Across all models, masking verbs causes substantially larger drops in spatial reasoning stability. $\Delta P_{SR}$ increases to 40–57% for OS-Genesis-7B, GUI-R1-7B, and AgentCPM-GUI-8B, while masking objects results in only minor changes in both $\Delta P_{Type}$ and $\Delta P_{SR}$. This indicates that agents rely on explicit action verbs as the anchor for reconstructing the intended instruction, forming a shortcut that bypasses textual reasoning.
>
> 2. In contrast, object tokens can often be inferred without significantly harming performance, suggesting they do not drive the model’s textual predictions.
>
> Overall, the verb token is the dominant determinant of $\Delta P_{SR}$ sensitivity. This shows that current GUI agents depend heavily on memorized verb–template patterns, directly explaining the observed sensitivity when verb cues are removed. The extend results and analysis are incorporated in our revised paper in Section 5.2.2 and Appendix B.3.2.
>
> > Q3: Comparisons with alternative process evaluators: Could Agent-ScanKit be combined with trajectory metrics (e.g., grounded-step checks, evidence-bank style) to see whether memory-heavy agents also show more ungrounded reflections?
>
> **Reply:** Thank you for your question. Owing to its benchmark-oriented design, Agent-ScanKit can be naturally extended to support trajectory-level process evaluation. As shown in Table Q3, we report trajectory-level task success rates (TSR) after masking in our vision-guided probing setup. The results show that, relative to the SFT model, agents trained with RL or CoT generally obtain lower TSR, suggesting stronger reflexive or exploratory behaviors. This trend is consistent with the findings from our step-level evaluation.
>
>
> **Table Q3: Trajectory-Level Metric: TSR**
> | Model               | TSR (%) |
> |---------------------|---------|
> | OS-ATLAS-Pro-7B     | 22.3    |
> | UI-TARS-7B-SFT      | 24.9    |
> | GUI-R1-7B           | 17.7    |
> | AgentCPM-GUI        | 14.7    |
> | GUI-Owl-7B          | 12.6    |
>
> Thus, Agent-ScanKit can operate at the trajectory level and can be viewed as complementary to existing grounded-step evaluators, while additionally exposing modality-specific vulnerabilities and memory-heavy behaviors.

---

> > ### Author Response · Authors · 2025-12-01
> >
> > > Q4: Practical guidance: Given your findings, what training interventions reduce shortcutting (e.g., layout-swap augmentation, coordinate jitter, instruction paraphrase curricula, reflection regularizers)—any preliminary results?
> >
> > **Reply:** Thank you for your question. We have conducted preliminary experiments to investigate the effectiveness of reflection regularization. The results are shown in Table Q4.
> >
> > **Table Q4: Reflection Regularization**
> >
> > ##### Supervised Fine-Tuning — Object Masking
> >
> > | GUI Agents          | Methods | ΔP_Type ↑ | ΔP_SR ↑ | VMC ↓ | RS ↑ |
> > |---------------------|---------|-----------|----------|--------|-------|
> > | **OS-ATLAS-Pro-7B** | Base    | 19.2 | 54.6 | 26.6 | 17.8 |
> > |                     | Prompt  | 24.2 | 57.0 | 25.1 | 22.7 |
> > | **OS-Genesis-7B**   | Base    | 0.99 | 2.03 | 96.1 | 0.39 |
> > |                     | Prompt  | 4.47 | 7.29 | 90.1 | 3.53 |
> > | **UI-TARS-7B-SFT**  | Base    | 1.40 | 28.4 | 64.8 | 0.99 |
> > |                     | Prompt  | 1.60 | 28.5 | 64.5 | 1.15 |
> >
> > ##### Reinforcement Learning — Object Masking
> >
> > | GUI Agents        | Methods | ΔP_Type ↑ | ΔP_SR ↑ | VMC ↓ | RS ↑ |
> > |-------------------|---------|-----------|----------|--------|-------|
> > | **GUI-R1-7B**     | Base    | 12.0 | 45.3 | 43.1 | 11.5 |
> > |                   | Prompt  | 17.4 | 47.6 | 40.7 | 16.0 |
> > | **AgentCPM-GUI-8B** | Base  | 29.6 | 51.7 | 33.2 | 22.8 |
> > |                   | Prompt  | 29.6 | 49.7 | 33.5 | 27.4 |
> > | **GUI-Owl-7B**    | Base    | 30.9 | 60.7 | 35.4 | 29.4 |
> > |                   | Prompt  | 39.1 | 67.1 | 30.5 | 36.7 |
> >
> > Our results show that existing GUI agents tend to fall back on memory-driven behaviors under perturbations, whereas lightweight reflective regularization effectively shifts them toward more deliberate, environment-grounded action selection. Furthermore, while SFT models benefit consistently across all metrics, RL/CoT agents exhibit model-specific gains. For example, reflective prompt suppress over-optimized shortcut policies in GUI-R1 and GUI-Owl, but offer limited improvement for AgentCPM-GUI due to its over-internalized prompting. Our results show that existing GUI agents tend to fall back on memory-driven behaviors under perturbations, whereas lightweight reflective regularization effectively shifts them toward more deliberate, environment-grounded action selection. Furthermore, while SFT models benefit consistently across all metrics, RL/CoT agents exhibit model-specific gains. For example, reflective prompt suppress over-optimized shortcut policies in GUI-R1 and GUI-Owl, but offer limited improvement for AgentCPM-GUI due to its over-internalized prompting. **More detailes can be found in Table 12 of Appendix B.2.7.**

---

### Official Review · Reviewer_B1cm · 2025-11-01

**Soundness:** 2
**Presentation:** 2
**Contribution:** 3
**Rating:** 4
**Confidence:** 4

**Summary:**

The paper introduces Agent‑ScanKit, a black‑box probing toolkit to disentangle memory vs reasoning in multimodal GUI agents via controlled visual, textual, and structural perturbations. It measures sensitivity by comparing performance with/without perturbations. Visual probes include object masking/editing (to test spatial memorization) and zoom‑in (to test local‑context reasoning); textual probes operate at token and sentence levels; structural probes diagnose visual and action shortcuts for reflection/state actions. Evaluated on 18 open models across 5 benchmarks, the study finds that models rely heavily on memorization and RL+CoT improves interpretability and some reasoning.

**Strengths:**

- The paper is well-motivated. The problem of distinguishing reasoning and memorization in GUI agents is important, and this work provides a timely response to it.

- The evaluation is comprehensive, as the study analyzes 18 open-source models across 5 benchmarks.

- The proposed approach of comparing performance with and without perturbations is innovative and reasonable, offering valuable insights into memorization behaviors in GUI agents.

**Weaknesses:**

- Although the analytical methods are innovative, the experimental design requires more rigor, and the current results appear to contain substantial noise.

  - In the visual-guided probing experiments, it is unclear whether the target elements are fully edited or masked. For example, in Figure 4, only the “Amazon App” icon is removed, while the text “Amazon” remains visible. Models could still take correct actions based on that textual cue, meaning this cannot be regarded as true “memorization.”

  - In the visual-guided probing case shown in Figure 4, a reasonable action would be to scroll left to find the “Amazon App” if the original app is masked. However, such actions are not included in the “RS”.

  - The analysis of “action shortcuts” seems less meaningful, as many actions (e.g., wait, complete, press back, press home) are naturally reasonable with only atomic guidance. For scroll, there are limited argument options. A high step-wise SR may not be very informative.

- Some assumptions and conclusions in the analysis appear overstated. For example, the sentence-level text probing results do not necessarily imply that the model relies on memorization. It could also indicate that the model is accustomed to following atomic commands or lacks sufficient planning ability. The authors should revisit their claims and provide more rigorous, evidence-backed interpretations in the whole paper.

- The paper needs thorough proofreading and improvements in presentation quality.

  - In Section 2, the phrase “This section reviews two lines of research that from the basis of this work” should use “form” instead of “from.”

  - The capitalization of subsection titles is inconsistent (e.g., Section 3.1 and Section 5.1).

  - The main paper should include brief explanations of the VMC and RS metrics, or at least reference the appendix where they are defined. Their absence makes it difficult for first-time readers to follow.

  - Tables 1 and 2 should explain the meaning of the arrows in $\Delta P$, VMC and RS. While it seems that the direction indicates better reasoning ability, this is not immediately clear to new readers.

  - The analysis sections are somewhat disorganized and would benefit from a clearer and more coherent logical structure.

- In paragraph 2 of Section 3.2, the discussion focuses on “weaknesses in training data hurting GUI agents,” but the connection to the concept of “infinite predictive space” is unclear and should be better articulated.

**Questions:**

Could the authors expand the analysis to closed-source models? It would be interesting to see how stronger proprietary models behave under similar perturbations.

---

> ### Author Response · Authors · 2025-11-27
>
> # Dear Reviewer B1cm
>
> **Thanks for your insightful review and constructive feedback.**
>
> > W1-1: In the visual-guided probing experiments, it is unclear whether the target elements are fully edited or masked. For example, in Figure 4, only the “Amazon App” icon is removed, while the text “Amazon” remains visible. Models could still take correct actions based on that textual cue, meaning this cannot be regarded as true “memorization.”
>
> **Reply:** Thank you for your suggerstion. To directly address this concern, we conducted an extensive sensitivity analysis by varying the masking/editing ratios from 10% to 100% across six representative models. We have updated the extended results and corresponding discussion in the paper (see Section 5.2.1 and Table 10 in Appendix B.2.5). We also provide partial results in Table W1-1. Our findings are as follows.
>
> **Table W1-1: Sensitivity analysis of masking and editing ratios in visual-guided probing.**
>
> | GUI Agents          | Ratio | Object Masking: ΔP_Type | Object Masking: ΔP_SR | Object Masking: VMC | Object Masking: RS | Object Editing: ΔP_Type | Object Editing: ΔP_SR | Object Editing: VMC | Object Editing: RS |
> |---------------------|-------|--------------------------|------------------------|----------------------|---------------------|--------------------------|------------------------|----------------------|---------------------|
> | **UI-TARS-7B-SFT**  | 10    | 8.30                     | 29.3                   | 64.8                 | 6.02                | 1.98                     | 6.10                   | 91.9                 | 1.00                |
> |                     | 30    | 9.40                     | 33.2                   | 59.3                 | 7.03                | 5.03                     | 20.7                   | 73.4                 | 3.30                |
> |                     | 50    | 10.7                     | 37.8                   | 53.3                 | 8.05                | 7.20                     | 33.8                   | 60.9                 | 5.12                |
> |                     | 70    | 12.3                     | 42.6                   | 46.9                 | 9.21                | 8.62                     | 41.5                   | 53.4                 | 6.12                |
> |                     | 100   | 14.8                     | 49.1                   | 39.5                 | 11.2                | 10.5                     | 46.7                   | 47.3                 | 7.53                |
> | **AgentCPM-GUI-8B** | 10    | 14.3                     | 35.8                   | 57.7                 | 13.6                | 1.73                     | 7.06                   | 92.9                 | 1.55                |
> |                     | 30    | 18.4                     | 44.0                   | 48.0                 | 17.6                | 8.59                     | 25.7                   | 69.2                 | 8.18                |
> |                     | 50    | 21.2                     | 49.5                   | 41.5                 | 20.4                | 13.2                     | 36.4                   | 56.0                 | 12.6                |
> |                     | 70    | 23.3                     | 45.3                   | 36.0                 | 22.4                | 15.7                     | 43.3                   | 48.3                 | 15.0                |
> |                     | 100   | 26.6                     | 60.9                   | 28.8                 | 25.6                | 18.9                     | 49.7                   | 42.2                 | 18.2                |
>
> 1. Even at 100% masking/editing, models still exhibit vsiual memory. Across models, VMC remains high (e.g., 9.87%$\sim$92.5% under masking; 21.6%$\sim$93.4% under editing), showing that models tend to click around the original location even when larger visual context is removed.
>
> 2. Scaling the perturbation ratio barely affects models with strong memory biases. Increasing the ratio from 10% to 100%, the model still exhibit a tendency towards coordinate predictions rather than bottom-up perception. Notably, the highest RS was only 25.6% for AgentCPM-GUI-8B at 100% masking rate.
>
> 3. RL/CoT models exhibit more robust reasoning than SFT models. Models such as AgentCPM-GUI and GUI-Owl exhabit larger changes of SR on heavy perturbations, highlithing an extent reflection that re-choose potential correct area. However, OS-Genesis remains nearly invariant across ratios, showing stronger postion-based memory. Notably, even without RL or CoT, OS-Atlas-Pro-7B exhibits sensitivity to perturbations, attributed to its prior learning of anomalies rather than overfitting to data.
>
> Overall, the supplementary results show that the models consistently rely on memory-driven inference, even under heavy masking or input edits.

---

> ### Author Response · Authors · 2025-11-27
>
> > W1-2: In the visual-guided probing case shown in Figure 4, a reasonable action would be to scroll left to find the “Amazon App” if the original app is masked. However, such actions are not included in the “RS”.
>
> **Reply:** Thank you for your suggerstion. To clarify, the RS metric explicitly covers reflective behaviors, including scrolling when the target cannot be visually located (e.g., ScrollLeft or ScrollRight). This definition was already used in all experiments and is provided in the appendix. To avoid ambiguity, we have now moved the complete RS definition into the main text (Page 6, visual-guided probing section). Thus, the action suggested by the reviewer is indeed counted within RS.
>
> > W1-3: The analysis of “action shortcuts” seems less meaningful, as many actions (e.g., wait, complete, press back, press home) are naturally reasonable with only atomic guidance. For scroll, there are limited argument options. A high step-wise SR may not be very informative.
>
> **Reply:** Thank you for your suggerstion. However, we suppose that this analysis is meaningful. To explain why current GUI agents maintain strong performance even without visual input, we introduce an intuitive ‘action shortcut’ analysis for reflection actions. These action types are usually ingnored in data and algorithms, generating shortcut learning. This, our result can reveal that their behavior is driven largely by low-level shortcut patterns rather than genuine multimodal reasoning. Our findings highlight three critical insights:
>
> 1. High SR under the action-shortcut setting indicates dependence on low-level priors instead of visual grounding. When visual input is removed, most agents still achieve high SR accuracy. This is not because these actions are genuinely “correct” in a multimodal sense, but because their training data builds strong shortcuts mapping between low-level instructions and predicted actions.
>
> 2. Such shortcuts lead to unreflective, rigid behavior. While these operations may appear reasonable in isolation, they reflect the model’s reliance on low-level instructions, which either inherited from the planner or user queries, rather than adaptive multimodal reasoning.
>
> 3. This highlights an important limitation: current agents behave more like instruction followers than perceptual AGI. Losing visual modality should have drastically reduced accuracy if models truly used visual grounding. Instead, performance remains high, revealing that their “competence” largely stems from atmoic instruction activation rather than genuine perception-action understanding.
>
> Thus, although these actions may seem intuitive, the analysis is meaningful because it uncovers a critical weakness that GUI agents succeed largely by exploiting memory shortcuts rather than by performing visually grounded decision-making.

---

> ### Author Response · Authors · 2025-11-27
>
> > W2: Some assumptions and conclusions in the analysis appear overstated. For example, the sentence-level text probing results do not necessarily imply that the model relies on memorization. It could also indicate that the model is accustomed to following atomic commands or lacks sufficient planning ability. The authors should revisit their claims and provide more rigorous, evidence-backed interpretations in the whole paper.
>
> **Reply:** Thank you for your thoughtful comments. We want to clarify our claims and conlcusion. First, most agents are optimized to follow step-wise, atomic instrcution rather than to reason over full trajectories. Thus, they learnt memory shortcuts that mapping atomic instructions to fixed outputs. Therefore, in real-world scenarios, as long as the atomic instructions change, the output will change accordingly, unaffected by visual input or high-level instructions. where agents trigger actions whenever a familiar atomic instruction appears, irrespective of visual and textual context, generting unreliable reasoning
>
> Our textual-guided probing further substantiates this phenomenon. At the token level, we show that masking key verb tokens almost completely eliminates the model’s ability to infer the textual content, demonstrating a brittle dependence on verb-anchored atomic templates (Table 2). In contrast, masking object tokens has negligible effect on prediction quality, indicating that the memory shortcut from action-content  instruction partern (Table 14 of Appendix B.3.2). At the sentence level, we observe that agents often execute incorrect or contradictory instructions even when these conflict with the visual state (Table 2), highlighting an over-compliance failure mode that further reflects the dominance of memorized atomic patterns over grounded reasoning.
>
> Overall, our work aims to provide a systematic probing analysis that exposes the fine-grained, memory-driven mechanisms underlying agents failures. By revealing how atomic instructions shape model inference at both sentence- and token-level granularity, our study helps clarify why current GUI agents struggle with state-aware. The revised content can be found in Section 5.2.2 and Appendix B.3.2.
>
> > W3: The paper needs thorough proofreading and improvements in presentation quality.
>
> **Reply:** Thank you for carefully reviewing our work. We have corrected the issues and improved the presentation quality in the revised version. Please refer to the blue-highlighted changes in the updated version. Specifically, the following revisions have been made:
>
> 1. Several typos have been corrected.
> 2. The capitalization of section headings has been standardized.
> 3. The arrows in Tables 1 and 2 now explicitly indicate higher levels of reasoning ability.
> 4. The analysis section has been reorganized to provide a clearer structure and a more coherent logical flow. For further details, please refer to Section 5.2.
>
> Overall, we have made every effort to improve the presentation quality within a short timeframe so that readers can better understand the contributions of this work.
>
> > W4: In paragraph 2 of Section 3.2, the discussion focuses on “weaknesses in training data hurting GUI agents,” but the connection to the concept of “infinite predictive space” is unclear and should be better articulated.
>
> **Reply:** Thank you for the reviewer’s insightful comment. We have revised such content in Section 3.2 to better articulate the connection between the training data and the concept of “infinite predictive space.”
>
> In the updated text (highlighted in blue), we explicitly clarify that GUI agents are simultaneously trained over two virtually unbounded predictive spaces—the coordinate space and the vocabulary space. Due to imbalanced data distributions and over-optimization within these infinite spaces, models develop memory bias and rely on shortcuts, which undermines their ability to learn genuine GUI-transition logic, especially for reflection and state actions. The revision also emphasizes that such shortcut behaviors directly contribute to SR degradation and mislead the evaluation of true reasoning capability.
>
> Overall, the revised paragraph now provides a clearer explanation of how training-data weaknesses interact with infinite predictive spaces, thereby sharpening the probing motivation for the following study.

---

> ### Author Response · Authors · 2025-11-27
>
> > Q1: Could the authors expand the analysis to closed-source models? It would be interesting to see how stronger proprietary models behave under similar perturbations.
>
> **Reply:** Thank you for your valuable suggestions. Incorporating closed-source models indeed provides a more comprehensive assessment of robustness under visual-guided perturbations. In the revised manuscript, we have expanded our analysis to include three proprietary multimodal agents—GPT-4o, GLM-4.5V, and Claude-4-Sonnet—evaluated under the same probing settings as the open-source models. Table Q1 reports their performance in visual-guieded probing and reveals two key observations:
>
> **Table Q1: Comparison of visual-guided probing on closed- and open-source multimodal agents in GUI tasks on the AITZ benchmark.**
>
> | GUI Agents                | Mask ΔP_Type ↑ | Mask ΔP_SR ↑ | Mask VMC ↓ | Mask RS ↑ | Edit ΔP_Type ↑ | Edit ΔP_SR ↑ | Edit VMC ↓ | Edit RS ↑ | Zoom ΔP_Type ↓ | Zoom ΔP_SR ↓ | Zoom VMC ↓ | Zoom RS ↓ |
> | ------------------------- | ---------------- | -------------- | ----------- | ---------- | ---------------- | -------------- | ----------- | ---------- | ---------------- | -------------- | ----------- | ---------- |
> | **GPT-4o**          | 9.07             | 34.4           | 44.3        | 1.74       | 8.44             | 35.3           | 42.1        | 1.40       | 15.8             | 99.0           | 38.2        | 0.36       |
> | **GLM-4.5V**        | 13.3             | 37.8           | 49.5        | 6.64       | 12.5             | 41.1           | 49.7        | 8.00       | 2.69             | 71.4           | 0.05        | 1.55       |
> | **Claude-4-Sonnet** | 14.1             | 40.4           | 40.4        | 2.17       | 1.00             | 31.4           | 53.3        | 0.99       | 1.77             | 97.2           | 14.1        | 1.21       |
> | **OS-ATLAS-Pro-7B** | 19.2             | 54.6           | 26.6        | 17.8       | 17.4             | 56.0           | 25.2        | 16.1       | 33.7             | 99.6           | 0.41        | 32.9       |
> | **OS-Genesis-7B**   | 0.99             | 2.03           | 96.1        | 0.39       | 0.73             | 2.76           | 96.4        | 0.52       | 9.13             | 99.6           | 73.4        | 5.09       |
> | **UI-TARS-7B-SFT**  | 1.40             | 28.4           | 64.8        | 0.99       | 1.88             | 33.7           | 58.5        | 1.19       | 1.32             | 52.0           | 5.04        | 0.51       |
> | **GUI-R1-7B**       | 12.0             | 45.3           | 43.1        | 11.5       | 11.6             | 45.3           | 40.8        | 10.6       | 1.63             | 61.4           | 2.37        | 1.35       |
> | **AgentCPM-GUI-8B** | 29.6             | 51.7           | 33.2        | 22.8       | 23.2             | 41.9           | 40.3        | 22.1       | 7.38             | 46.4           | 1.52        | 7.11       |
> | **GUI-Owl-7B**      | 30.9             | 60.7           | 35.4        | 29.4       | 29.3             | 58.1           | 36.6        | 27.7       | 8.02             | 30.9           | 1.68        | 7.53       |
>
>
> 1. Among closed source systems, newer models such as GLM-4.5V, which integrate GUI oriented capabilities, show higher RS under memory probing and robust reasoning under Zoom in probing. In contrast, GPT-4o shows unreliable when object are masked or edited and undergoes a clear reasoning collapse under Zoom in probing.
> 2. Furthermore, the overall robustness of closed source models stays broadly comparable to SFT based open source models. Within the SFT group, OS-ATLAS-Pro-7B shows desirable anomaly sensitivity, whereas RL trained agents such as AgentCPM-GUI-8B and GUI-Owl-7B are even more responsive to perturbations and tend to maintain relatively stable reasoning under Zoom-in setting.
>
> Thus, general purpose multimodal models remain unreliable when performing GUI tasks, fail to detect visual anomalies, and are unable to guarantee reliable inference in local scenarios. The extended results and analysis are revised in our paper (Section 5.2.1 and Appendix~\ref{B.2.1}). Furthermore, we also report the result of closed-source agents for the textual-guided probing and structure-guided probing. Our findings are as follows:
>
> 1. **Textual-guided probing:** as with open source agents, closed-source agents also rely on complete atomic instructions and frequently over-execution with misleading instructions (Appendix B.3.1).
>
> 2. **Structure-guided probing:** Closed-source models rely on action shortcuts yet preserve strong visual reasoning, whereas open-source agents exhibit polarized shortcut use and weaker visual robustness (B.4.1).

---

### Author Response · Authors · 2025-11-27
**General Purpose**

**Dear reviewers and AC,**

We are encouraged that the reviewers recognize our work as well-motivated (B1cm, ct7c, f44Q); as a comprehensive evaluation (B1cm, xwUw); as innovative and reasonable (B1cm, ct7c, xwUw, f44Q); as high quality (ct7c); and as offering an insightful direction for future study (xwUw).

We also thank all the reviewers for their insightful comments. Their concerns and suggestions focus on sensitivity and ablation studies (B1cm, xwUw, ct7c), motivation and purpose (B1cm, f44Q), and presentation (B1cm, f44Q). We revise our paper accordingly, provide individual responses to each reviewer, and incorporate the reviewers suggestions into the revised version highlighted in blue. The main changes to the paper are summarized as follows.

> **Reviewer B1cm:**

**W1-1:** We add sensitivity analysis of masking/editing ratio for visual-guided probing (Table 10 of Appendix B.2.5).

**W1-2:** We explain the definitions and implications of the metrics (Line 312-316, Page 6 and Appendix A.5).

**W1-3:** We highlight the importance of action shortcuts in probing (Section 4.3, Section 5.2.3, and Fig.12 of Appendix B.4.2).

**W2:** We clarify the effect of atomic instructions from a reasoning/memory perspective, and add more analysis for atomic instructions in textual-guided probing (Section 5.2.2 and Table 14 of Appendix B.3.2).

**W3:** We improve presentation quality in the revised paper.

**W4:** We strengthen the relationship between the infinite predictive space and the effect of training data hurting in Section 3.2.

**Q1:** We add closed-source model results and analyses across the three probing paradigms (Section 5.2.1; Appendix Tables 8, 13, 15).

> **Reviewer ct7c:**

**W1:** We add masking/editing ratios, zoom-in ratios, text-modal ablation to enhance causal interpretation of reasoning and memorization (Table 10-11 of Appendix B.2.5), includ non-target masking/editing results as ablation controls (Table 9 of Appendix B.2.4), and clarify the purpose of the three probing methods (Section 4.1~4.3).

**W2:** We add sensitivity sweeps for VMC thresholds (Fig. 9 of Appendix B.2.5), clarify RS (Line 312-316, Page 6), and provide analysis of RS in Section 5.2.1 and Appendix B.2.

**W3:** We clarify the definition of structural probes in Section 4.3 and Section 5.2.3, and add scale changes (zoom ratios) in the Table 11 of Appendix B.2.5 and AITZ theme analysis in the Table 15 of Appendix B.4.1.

**Questions:**
- Q1: We provide analysis through zoom-in ratios and clarified the target decision support after perturbation in the Table 11 of Appendix B.2.5.
- Q2: We extend token-level analysis in the Table 14 of Appendix B.3.2.
- Q3: We add trajectory-level metric outcomes in rebuttal reply.
- Q4: We add mitigation strategy in the Table 12 of Appendix B.2.7.



> **Reviewer xwUw:**

**W1:** We provide sensitivity analysis of masking/editing ratios (Table 10 of Appendix B.2.5) to capture nuance of memory and reasoning, clarify that zoom-in serves as a reasoning-focused probing method (Section 4.1) and add sensitivity analysis of zoom-in ratios results for further reasoning evaluations  (Table 11 of Appendix B.2.5) .

**W2&Q1:** We add text-modal ablation under visual probing for the dependency on visual memory change when the textual input is removed or perturbed first (Fig.7 of Appendix B.2.3).

**W3:** We explain why RL still struggles with memory bias, including global visual-structure memory in zoom-in probing and reward hijacking that leads to instruction-following and action shortcuts.

**Questions:**
- Q2: We evaluate token cost for each CoT-based models, explained the shortcomings of RL
- Q3: We interprete the OS-Genesis results.

> **Reviewer f44Q:**

**W1:** We clarify the purpose of three probing frameworks to improve readability (Section 4.1-4.3), explain interpretation of the four core metrics (Section 4, Appendix A.5), and improve presentation linking data to core concepts (Section 5.2.1-5.2.3).

**W2:** To strengthen the conclusion that textual-guided probing reveals memory-driven behavior, we add token-level analyses of verbs and objects to illustrate internal token-level memory (Appendix B.3.2, Table 14). We also highlight that sentence-level memory arises from the fixed mapping between error-atomic instructions and outputs in Section 5.2.2. Moreover, action shortcuts further indicate that agents often make decisions without relying on visual input in Section 5.2.3.

**Qestions:**
- Q1: We provide a case study to explain the implementations of visual- and actions- shortcuts (Q1, Fig.12 of Appendix B.4.2),
- Q2: We clarify the quantitative metrics in Section 4.1 and Appendix A.5, explain the definition of over-reflection in RS (Section 5.2.1), revise the formal definition of genuine reasoning in zoom-in analysis (Section 4.1), and improve the perturbed POMDP framework for better understanding.

We thank the Area Chair and reviewers again for their time and effort.

---

### Meta-Review · Area_Chair_uz8w · 2026-01-07

**Summary:**

The paper proposes a probing toolkit that uses controlled visual, textual, and structural perturbations to separate memorization from genuine reasoning in multimodal GUI agents, and the angle is timely and genuinely insightful for understanding why agents look competent but fail out of distribution. However, the core causal claim (delta under perturbation = memory vs. reasoning) still feels under-justified and noisy, several probes admit confounds, and the metric/threshold choices and “shortcut” interpretations aren’t fully convincing despite added ablations. I would recommend borderline rejection.

**Reviewer Concerns:**

- Rebuttal addressed: (1) added sensitivity/ablation studies for visual probes (mask/edit ratios...), (2) clarified key metrics/threshold sensitivity and improved presentation, (3) added closed-source model results plus extra token-level/trajectory-level analyses.

- Still outstanding: (1) causal interpretation remains indirect—delta metrics can still conflate missing-context effects with “memorization vs. reasoning”, (2) structural probing scope is limited, (3) core claims remain stronger than the evidence, with important support living mainly in the appendix rather than the main paper.

**Reviewer Scores:**

- B1cm: a bit more positive, since masking/editing sensitivity, RS definition, and closed-source comparisons directly address their main rigor/clarity concerns.
- ct7c: likely unchanged, as the “delta under perturbation ⇒ memory vs. reasoning” causal framing still has confounds even with added ablations/sweeps.
- xwUw: a bit more positive, because the added modality ablations and clearer separation of “memory probes” vs. “OOD zoom-in” reduce their key ambiguity.
- f44Q: likely unchanged, given mainly improved readability.

---

### Decision · Program_Chairs · 2026-01-26

Reject